# HQ-VAE: Hierarchical Discrete Representation Learning with Variational Bayes

## Abstract

Vector quantization (VQ) is a technique to deterministically learn features with discrete codebook representations. It is commonly achieved with a variational autoencoding model, VQ-VAE, which is further extended to hierarchical structures for high-fidelity reconstruction. However, hierarchical extensions of VQ-VAE often suffer from codebook/layer collapse issue, where the codebook is not efficiently used to express data well, hence deteriorates reconstruction accuracy. To mitigate this problem of the extensions, we propose a novel unified framework to stochastically learn hierarchical discrete representation on the basis of the variational Bayes framework, called hierarchically quantized variational autoencoder (HQ-VAE). HQ-VAE naturally generalizes the hierarchical variants of VQ-VAE such as VQ-VAE-2 and residual-quantized VAE (RQ-VAE) and provides them with a Bayesian training scheme. Our comprehensive experiments on image datasets show that HQ-VAE enhances codebook usage and improves reconstruction performance. We also validate HQ-VAE in terms of its applicability even to a different modality with an audio dataset.

## 1 Introduction

Learning representations with discrete features is one of the core technologies in the field of deep learning. Vector quantization (VQ) for approximating continuous features with a set of finite trainable code vectors is a common technique to achieve such representation (Toderici et al., 2016; Theis et al., 2017; Agustsson et al., 2017). It has been widely adopted in several active applications including neural codecs, e.g., image compression (Williams et al., 2020; Wang et al., 2022) and audio codec (Zeghidour et al., 2021; Défossez et al., 2022). VQ-based representation methods have been improved with successful deep generative modeling, especially denoising diffusion probabilistic models (Sohl-Dickstein et al., 2015; Ho et al., 2020; Song et al., 2020; Dhariwal & Nichol, 2021; Hoogeboom et al., 2021; Austin et al., 2021) and autoregressive models (van den Oord et al., 2016; Chen et al., 2018; Child et al., 2019). Learning discrete features of target data among finitely many representations can ignore redundant information, and such a lossy compression can assist with training deep generative models on large-scale data. After compression, one can train another deep generative model, which is called a prior model, on the compressed representation instead of the raw data. This approach has achieved attractive results in various tasks, e.g., unconditional generation tasks (Razavi et al., 2019; Dhariwal et al., 2020; Esser et al., 2021b;a; Rombach et al., 2022), text-to-image generation (Ramesh et al., 2021; Gu et al., 2022; Lee et al., 2022a) and textually guided audio generation (Yang et al., 2022; Kreuk et al., 2022). Note that the compression performance of VQ limits the overall generation performance regardless of the performance of the prior model.

VQ is usually achieved with the model vector quantized variational autoencoder (VQ-VAE) (van den Oord et al., 2017). In VQ-VAE, an input is first encoded and quantized with the code vectors, which extracts the discrete representation of the encoded feature. The discrete representation is then decoded to the data space to recover the original input. Subsequently, advanced studies incorporated the hierarchical structure into the discrete latent space to effectively achieve high-fidelity reconstruction. Razavi et al. (2019) initially extended VQ-VAE to a hierarchical model, which is called VQ-VAE-2. In this model, multi-resolution discrete latent representations are introduced to extract local and global information of the target data. As another type

of hierarchical discrete representation, residual quantization (RQ), was proposed to reduce the gap between the feature maps before and after the quantization process (Zeghidour et al., 2021; Lee et al., 2022a).

Despite its successes in many tasks, training variants of VQ-VAE is still challenging. It is known that VQ-VAE suffers from codebook collapse, where only few code vectors are used for the representation (Kaiser et al., 2018; Roy et al., 2018; Takida et al., 2022b). This inefficiency may deteriorate reconstruction accuracy, hence, limit its applications to downstream tasks. The extension with hierarchical latent representations suffers from the same issue. For example, Dhariwal et al. (2020) reported that it is generally difficult to push information to higher levels in VQ-VAE-2, i.e., codebook collapse often occurs there. Therefore, certain heuristic techniques such as the exponential moving average (EMA) update (Polyak & Juditsky, 1992) and codebook reset (Dhariwal et al., 2020) are usually implemented to mitigate these problems. Takida et al. (2022b) claimed that the issue is triggered because the training scheme of VQ-VAE does not follow the variational Bayes framework but relies on carefully designed heuristics. They proposed stochastically quantized VAE (SQ-VAE), with which the components of VQ-VAE, i.e., the encoder, decoder and code vectors, are trained in the variational Bayes framework with an SQ operator. The model was shown to improve reconstruction performance by preventing the collapse issue thanks to the *self-annealing* effect (Takida et al., 2022b), where the SQ process gradually tends to the deterministic one during training. We expect this has the potential to stabilize the training by mitigating this problem even in the hierarchical model, which may lead to improving reconstruction performance with more efficient codebook usage.

We propose *Hierarchically Quantized VAE* (*HQ-VAE*), a general variational Bayesian model for learning hierarchical discrete latent representations. Figure 1 illustrates the overall architecture of HQ-VAE. The hierarchical structure in HQ-VAE consists of *bottom-up* and *top-down* path pair, which assists with capturing local and global information of data. We instantiate the generic HQ-VAE by introducing two types of *top-down* layers. These two layers formulate the hierarchical structures of VQ-VAE-2 and residual-quantized VAE (RQ-VAE) within the variational scheme, which we call *SQ-VAE-2* and *RSQ-VAE*, respectively. In other words, our framework can deal with the independently proposed extensions of VQ-VAE in a unified and solid way. HQ-VAE can be viewed as an extension of SQ-VAE with hierarchy, hence, shares similar favorable properties of SQ-VAE (e.g., the *self-annealing* effect). In this sense, HQ-VAE unifies the current well-known VQ models in the variational Bayes framework, which provides a novel training mechanism. We empirically show HQ-VAE improves upon conventional methods in the vision and audio domains. Furthermore, through the demonstration of the hybrid model of SQ-VAE-2 and RSQ-VAE (in Appendices B and C.3), we provide a tutorial for modeling hierarchical discrete representation from the design of hierarchical latent structures to the derivation of objective functions to learn the model. The demonstration shows the flexibility of our framework, which allows to model desired hierarchical discrete latent via the stack of *top-down* layers. This paper is the first attempt at variational Bayes on hierarchical discrete representation, and training hierarchical VAEs is generally known to be challenging even for continuous latent cases (Vahdat & Kautz, 2020; Child, 2021).

Throughout this paper, the uppercase letters ($P$, $Q$) and the lowercase letters ($p$, $q$) denote the probability mass functions and probability density functions, respectively; calligraphy letters ($\mathcal{P}$, $\mathcal{Q}$) denote the joint probabilistic distributions of both continuous and discrete random variables; bold lowercase and uppercase letters (e.g., $\boldsymbol{x}$ and $\boldsymbol{Y}$) respectively denote vectors and matrices, and the $i$th column vector in $\boldsymbol{Y}$ is written as $\boldsymbol{y}_i$; $[N]$ denotes a set of positive integers no more than $N \in \mathbb{N}$. Finally, we use $\mathcal{J}$ and $\mathcal{L}$ for the objective functions of HQ-VAE and conventional ones, respectively.

## 2 Background

We first revisit VQ-VAE and its extensions to hierarchical latent models. We then review SQ-VAE which serves as the foundation framework of HQ-VAE.

### 2.1 VQ-VAE

To discretely represent observations $\boldsymbol{x} \in \mathbb{R}^D$, a codebook $\boldsymbol{B}$ is introduced, which consists of finite trainable code vectors $\{\boldsymbol{b}_k\}_{k=1}^K$ ($\boldsymbol{b}_k \in \mathbb{R}^{d_b}$). A discrete latent variable $\boldsymbol{Z}$ is constructed to be in the $d_z$-tuple of $\boldsymbol{B}$,

i.e., $\boldsymbol{Z} \in \boldsymbol{B}^{d_z}$, which is later decoded to generate data samples. To connect the observation and latent representaion, a deterministic encoder and decoder pair is introduced, where the encoder maps $\boldsymbol{x}$ to $\boldsymbol{Z}$ and the decoder recovers $\boldsymbol{x}$ from $\boldsymbol{Z}$ by a decoding function $\boldsymbol{f_\theta} : \mathbb{R}^{d_b \times d_z} \to \mathbb{R}^D$. For the encoder, an encoding function, denoted as $\boldsymbol{G_\phi} : \mathbb{R}^D \to \mathbb{R}^{d_b \times d_z}$, and a deterministic quantization operator are introduced. The encoding function first maps $\boldsymbol{x}$ to $\hat{\boldsymbol{Z}} \in \mathbb{R}^{d_b \times d_z}$, then the quantization operator finds the nearest neighbor of $\hat{\boldsymbol{z}}_i$ for $i \in [d_z]$, i.e., $\boldsymbol{z}_i = \arg\min_{\boldsymbol{b}_k} \|\hat{\boldsymbol{z}}_i - \boldsymbol{b}_k\|_2^2$. The trainable components (the encoder, decoder, and codebook) are learned by minimizing the objective

$$\mathcal{L}_{\text{VQ-VAE}} = \|\boldsymbol{x} - \boldsymbol{f_\theta}(\boldsymbol{Z})\|_2^2 + \beta\|\hat{\boldsymbol{Z}} - \text{sg}[\boldsymbol{Z}]\|_F^2, \tag{1}$$

where $\text{sg}[\cdot]$ is the stop-gradient operator and $\beta$ is a hyperparameter balancing the two terms. The codebook is updated by applying the EMA update to $\|\text{sg}[\hat{\boldsymbol{Z}}] - \boldsymbol{Z}\|_F^2$.

**VQ-VAE-2.** To model both local and global information separately, VQ-VAE-2 adopts a hierarchical structure for vector quantization (Razavi et al., 2019). The model consists of multiple levels of latents so that top levels have global information while bottom levels are focused on local information, conditioned on the top levels. The training of the model follows the same scheme as the original VQ-VAE (e.g., stop-gradient, the EMA update, and deterministic quantization).

**RQ-VAE.** RQ provides a finer approximation of $\boldsymbol{Z}$ by taking into account the information of quantization gaps (residuals) (Zeghidour et al., 2021; Lee et al., 2022a). With RQ, $L$ code vectors are assigned to each vector $\boldsymbol{z}_i$ ($i \in [d_z]$), instead of increasing the codebook size $K$. To achieve multiple assignments, RQ repeatedly quantizes the target feature and computes quantization residuals, denoted as $\boldsymbol{R}_l$. Namely, the following procedure is repeated $L$ times starting with $\boldsymbol{R}_0 = \hat{\boldsymbol{Z}}$: $\boldsymbol{z}_{l,i} = \arg\min_{\boldsymbol{b}_k} \|\boldsymbol{r}_{l-1,i} - \boldsymbol{b}_k\|_2^2$ and $\boldsymbol{R}_l = \boldsymbol{R}_{l-1} - \boldsymbol{Z}_l$. By repeating RQ, the discrete representation is expected to be refined in a coarse-to-fine manner. Finally, RQ discretely approximates the encoded variable as $\hat{\boldsymbol{Z}} \approx \sum_{l=1}^L \boldsymbol{Z}_l$, where the conventional VQ is regarded as a special case of RQ with $L = 1$.

## 2.2 SQ-VAE

SQ-VAE (Takida et al., 2022b) also has deterministic encoding/decoding functions and a trainable codebook. However, unlike the deterministic quantization scheme of VQ and RQ, SQ-VAE designs an SQ procedure for the encoded features following the variational Bayes framework. More precisely, it defines a stochastic dequantization process $p_{s^2}(\tilde{\boldsymbol{z}}_i|\boldsymbol{Z}) = \mathcal{N}(\tilde{\boldsymbol{z}}_i; \boldsymbol{z}_i, s^2\boldsymbol{I})$, which converts a discrete variable $\boldsymbol{z}_i$ into a continuous one $\tilde{\boldsymbol{z}}_i$ by adding Gaussian noise with a learnable variance $s^2$. By Bayes' rule, it associates with a reverse operation, i.e., SQ, which is given by $\hat{P}_{s^2}(\boldsymbol{z}_i = \boldsymbol{b}_k|\tilde{\boldsymbol{Z}}) \propto \exp\left(-\frac{\|\tilde{\boldsymbol{z}}_i - \boldsymbol{b}_k\|_2^2}{2s^2}\right)$. Thanks to this variational framework, the degree of the stochasticity in the quantization scheme becomes adaptive. This allows SQ-VAE to benefit from the effect of *self-annealing*, where the SQ process gradually approaches the deterministic one as $s^2$ decreases. This generally improves the efficiency of codebook usage.

**SQ-VAE vs. dVAE.** Sønderby et al. (2017) proposed discrete latent VAE with a codebook lookup. Ramesh et al. (2021) followed the same approach to train VAE equipped with a codebook for text-to-image generation and call the model discrete VAE (dVAE). In dVAE, stochastic posterior categorical distribution for which code vectors are assigned is directly modeled by the encoder as $Q(\boldsymbol{z}_i = \boldsymbol{b}_k|\boldsymbol{x}) \propto \exp(g_{\phi,k}(\boldsymbol{x}))$, where $\boldsymbol{g_\phi} : \mathbb{R}^D \to \mathbb{R}^K$. This modeling enables to encode an original sample $\boldsymbol{x}$ into a set of codes $\boldsymbol{Z}$ in a probabilistic way. However, such index-domain modeling cannot incorporate the codebook geometry explicitly into the posterior modeling. In contrast, VQ/SQ-based methods allow us to model the posterior distribution with vector operations such as VQ and RQ. Furthermore, as another benefit of VQ/SQ-VAE, it enables to evaluate the reconstruction errors coming from the discretization with the quantization errors (Dhariwal et al., 2020). In this paper, we restrict our scope to latent generative model involving vector quantization operators.

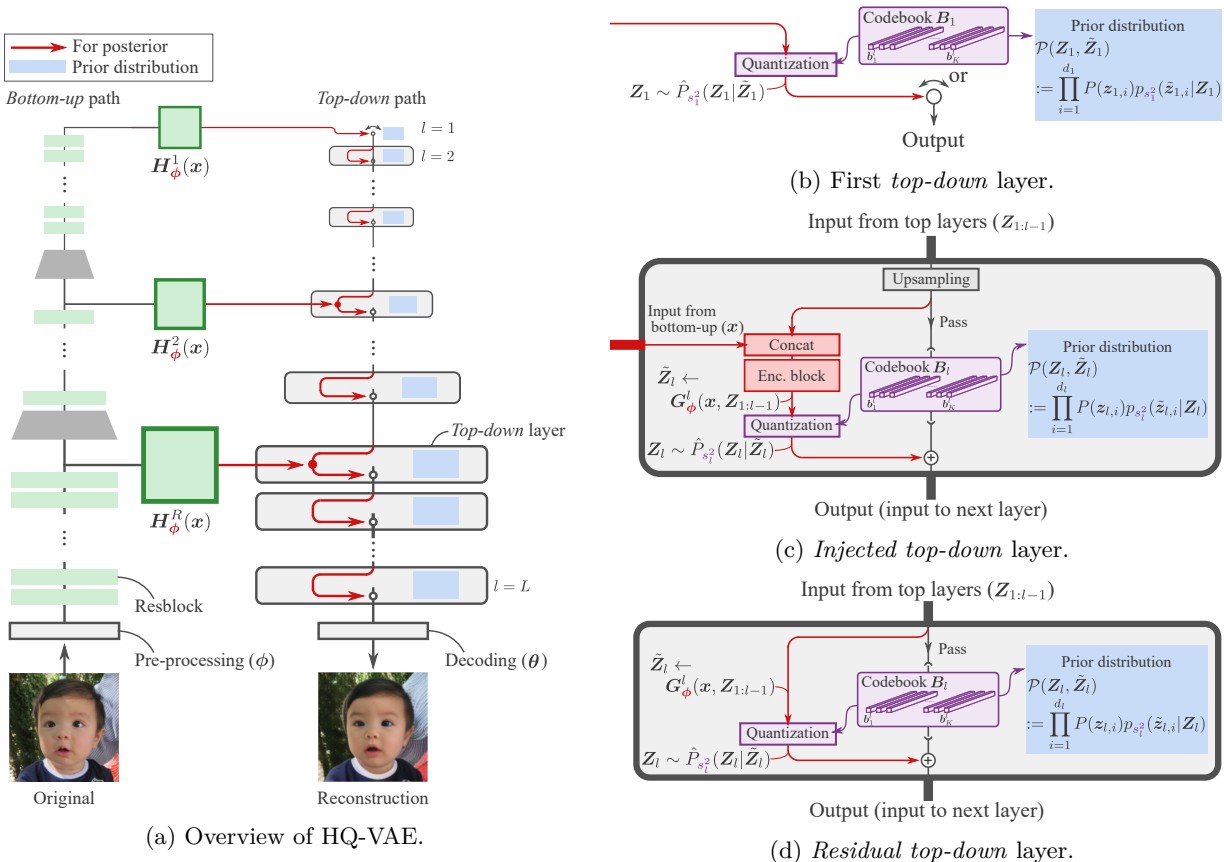

Figure 1: (a) HQ-VAE consists of *bottom-up* and *top-down* paths. Red arrows are for approximated posterior. Kullback–Leibler divergence of posterior and prior (in the blue box) is evaluated for objective function. (b) First layer for *top-down* path. (c)-(d) We introduce two types of layers: *injected top-down* and *residual top-down*. HQ-VAE that consists only of the injected (residual) *top-down* layer is analogous to VQ-VAE-2 (RQ-VAE).

# 3 Hierarchically quantized VAE

In this section, we formulate the generic HQ-VAE model, which learns hierarchical discrete latent representation in the variational Bayes framework. It serves as a backbone of the instantiations of HQ-VAE presented in Section 4.

To achieve hierarchical discrete representation of depth $L$, we first introduce $L$ groups of discrete latent variables, which are denoted as $\boldsymbol{Z}_{1:L} := \{\boldsymbol{Z}_l\}_{l=1}^L$. For each $l \in [L]$ we introduce a trainable codebook $\boldsymbol{B}_l := \{\boldsymbol{b}_k^l\}_{k=1}^{K_l}$, consisting of $K_l$ $d_b$-dimensional code vectors, i.e., $\boldsymbol{b}_k^l \in \mathbb{R}^{d_b}$ for $k \in [K_l]$. The variable $\boldsymbol{Z}_l$ is represented as a $d_l$-tuple of the code vectors in $\boldsymbol{B}_l$; namely, $\boldsymbol{Z}_l \in \boldsymbol{B}_l^{d_l}$. Similarly to conventional VAEs, the latent variable of each group is assumed to follow a pre-defined prior mass function. We set the prior as an i.i.d. uniform distribution, defined as $P(\boldsymbol{z}_{l,i} = \boldsymbol{b}_k) = 1/K_l$ for $i \in [d_l]$. The probabilistic decoder is set as a normal distribution with a trainable isotropic covariance matrix as $p_{\boldsymbol{\theta}}(\boldsymbol{x}|\boldsymbol{Z}_{1:L}) = \mathcal{N}(\boldsymbol{x}; \boldsymbol{f}_{\boldsymbol{\theta}}(\boldsymbol{Z}_{1:L}), \sigma^2 \boldsymbol{I})$ with a decoding function $\boldsymbol{f}_{\boldsymbol{\theta}} : \mathbb{R}^{d_b \times d_1} \oplus \cdots \oplus \mathbb{R}^{d_b \times d_L} \to \mathbb{R}^D$. It decodes latent variables sampled from the prior to generate instances. Here, the exact evaluation of $P_{\boldsymbol{\theta}}(\boldsymbol{Z}_{1:L}|\boldsymbol{x})$ is required to train the generative model with the maximum likelihood. However, it is intractable in practice. Thus, we introduce an approximated posterior on $\boldsymbol{Z}_{1:L}$ given $\boldsymbol{x}$ and derive the evidence lower bound (ELBO) for maximization instead.

Inspired by hierarchical Gaussian VAEs (Sønderby et al., 2016; Vahdat & Kautz, 2020; Child, 2021), HQ-VAE consists of *bottom-up* and *top-down* paths, as shown in Figure 1a. The approximated posterior has the *top-down* structure ($\boldsymbol{Z}_1 \to \boldsymbol{Z}_2 \to \cdots \to \boldsymbol{Z}_L$). For this process, the *bottom-up* path first generates features from $\boldsymbol{x}$ as $\boldsymbol{H}_{\boldsymbol{\phi}}^r(\boldsymbol{x})$ at different resolutions ($r \in [R]$). In the *top-down* path, the latent variable in each group is processed in the order from $\boldsymbol{Z}_1$ to $\boldsymbol{Z}_L$ by taking $\boldsymbol{H}_{\boldsymbol{\phi}}^r(\boldsymbol{x})$ into account. To achieve this, two features including that extracted by the *bottom-up* path ($\boldsymbol{H}_{\boldsymbol{\phi}}^r(\boldsymbol{x})$) and that processed at higher layers in the *top-down* path ($\boldsymbol{Z}_{1:l-1}$) can be fed to each layer and processed to estimate $\boldsymbol{Z}_l$ corresponding to $\boldsymbol{x}$, which we denote it as $\hat{\boldsymbol{Z}}_l = \boldsymbol{G}_{\boldsymbol{\phi}}^l(\boldsymbol{H}_{\boldsymbol{\phi}}^r(\boldsymbol{x}), \boldsymbol{Z}_{1:l-1})$. The $l$th group $\boldsymbol{Z}_l$ has a unique resolution index $r$, and we denote it as $r(l)$. For simplicity, we ignore $\boldsymbol{H}_{\boldsymbol{\phi}}^r$ in $\hat{\boldsymbol{Z}}_l$ and write $\hat{\boldsymbol{Z}}_l = \boldsymbol{G}_{\boldsymbol{\phi}}^l(\boldsymbol{x}, \boldsymbol{Z}_{1:l-1})$. The design of the encoding function $\boldsymbol{G}_{\boldsymbol{\phi}}^l$ brings us to different modeling of the approximated posterior. We leave the detailed discussion in the next section.

It should be noted that the outputs of $\boldsymbol{G}_{\boldsymbol{\phi}}^l$ lie in $\mathbb{R}^{d_z \times d_b}$, whereas the support of $\boldsymbol{Z}_l$ is restricted to $\boldsymbol{B}_l^{d_z}$. To connect these continuous and discrete spaces, we introduce a pair of stochastic dequantization and quantization processes, as in Takida et al. (2022b). We first define the stochastic dequantization process for each group as

$$p_{s_l^2}(\tilde{\boldsymbol{z}}_{l,i}|\boldsymbol{Z}_l) = \mathcal{N}(\tilde{\boldsymbol{z}}_{l,i}; \boldsymbol{z}_{l,i}, s_l^2 \boldsymbol{I}), \tag{2}$$

which is equivalent to adding Gaussian noise to the discrete variable the covariance of which, $s_l^2 \boldsymbol{I}$, depends on the index of the group $l$. We hereafter denote the set of $\tilde{\boldsymbol{Z}}_l$ as $\tilde{\boldsymbol{Z}}_{1:L}$, i.e., $\tilde{\boldsymbol{Z}}_{1:L} := \{\tilde{\boldsymbol{Z}}_l\}_{l=1}^L$. Next, we can derive a stochastic quantization process as the inverse operator of the above stochastic dequantization:

$$\hat{P}_{s_l^2}(\boldsymbol{z}_{l,i} = \boldsymbol{b}_k|\tilde{\boldsymbol{Z}}_l) \propto \exp\left(-\frac{\|\tilde{\boldsymbol{z}}_{l,i} - \boldsymbol{b}_k\|_2^2}{2s_l^2}\right). \tag{3}$$

By using these stochastic operators, we can connect $\hat{\boldsymbol{Z}}_{1:L}$ and $\boldsymbol{Z}_{1:L}$ via $\tilde{\boldsymbol{Z}}_{1:L}$ in a stochastic manner, which leads to the entire encoding process:

$$\mathcal{Q}(\boldsymbol{Z}_{1:L}, \tilde{\boldsymbol{Z}}_{1:L}|\boldsymbol{x}) = \prod_{l=1}^L \prod_{i=1}^{d_l} p_{s_l^2}(\tilde{\boldsymbol{z}}_{l,i}|\boldsymbol{G}_{\boldsymbol{\phi}}^l(\boldsymbol{x}, \boldsymbol{Z}_{1:l-1})) \hat{P}_{s_l^2}(\boldsymbol{z}_{l,i}|\tilde{\boldsymbol{Z}}_l). \tag{4}$$

The prior distribution on $\boldsymbol{Z}_{1:L}$ and $\tilde{\boldsymbol{Z}}_{1:L}$ is defined using the stochastic dequantization process as

$$\mathcal{P}(\boldsymbol{Z}_{1:L}, \tilde{\boldsymbol{Z}}_{1:L}) = \prod_{l=1}^L \prod_{i=1}^{d_l} P(\boldsymbol{z}_{l,i}) p_{s_l^2}(\tilde{\boldsymbol{z}}_i|\boldsymbol{Z}_l), \tag{5}$$

where the latent representations are generated in the order from $l = 1$ to $L$. The generative process from the prior does not use $\tilde{\boldsymbol{Z}}$ but $\boldsymbol{Z}$ as $\boldsymbol{x} = \boldsymbol{f}_{\boldsymbol{\theta}}(\boldsymbol{Z})$.

# 4 Instantiations of HQ-VAE

Now that we have established the overall framework of HQ-VAE, we consider two special cases of HQ-VAE by designing two types of *top-down* layers: *injected top-down* and *residual top-down*. We derive two instances of HQ-VAE that consists only of the *injected top-down* layer or the *residual top-down* layer, which we call SQ-VAE-2 and RSQ-VAE, respectively due to their analogue to VQ-VAE-2 and RQ-VAE. These two layers can be combinatorially used to define a hybrid model of SQ-VAE-2 and RSQ-VAE, which is explained in Appendix B. Note that the prior distribution (Equation (5)) is identical across all instantiations.

## 4.1 First top-down layer

We introduce the first *top-down* layer, which is put at the top of layers in HQ-VAE. As illustrated in Figure 1b, this layer takes $\boldsymbol{H}_{\boldsymbol{\phi}}^1(\boldsymbol{x})$ as an input and processes it with SQ. HQ-VAE constructed only with this layer reduces to SQ-VAE.

## 4.2 Injected top-down layer

We design an *injected top-down* layer for the approximated posterior as in Figure 1c. This layer infuses the variable processed in the *top-down* path with the higher resolution information from the *bottom-up* layer. The $l$th layer takes the feature from the *bottom-up* path ($\boldsymbol{H}_{\boldsymbol{\phi}}^{r(l)}(\boldsymbol{x})$) and the variable from the higher groups in the *top-down* path as inputs. In the layer, the variable from higher layers is first upsampled to be aligned with $\boldsymbol{H}_{\boldsymbol{\phi}}^{r(l)}(\boldsymbol{x})$. These two variables are then concatenated and processed with an encoding block. The above overall process corresponds to $\hat{\boldsymbol{Z}}_l = \boldsymbol{G}_{\boldsymbol{\phi}}^l(\boldsymbol{x}, \boldsymbol{Z}_{1:l-1})$ in Section 3. The encoded variable $\hat{\boldsymbol{Z}}_l$ is then quantized into $\boldsymbol{Z}_l$ with the codebook $\boldsymbol{B}_l$ through the process described in Equation (3)[1]. Finally, the sum of the variable from the top layers and quantized variable $\boldsymbol{Z}_l$ is passed through to the next layer.

### 4.2.1 SQ-VAE-2

We especially instantiate the HQ-VAE only with the *injected top-down* layers in addition to the first layer, which reduces to SQ-VAE-2. Note that since the index of resolutions and layers have a one-to-one correspondence in this structure, $r(l) = l$ and $L = R$. As in usual VAEs, we evaluate the ELBO as $\log p_{\boldsymbol{\theta}}(\boldsymbol{x}) \geq -\mathcal{J}_{\text{SQ-VAE-2}}(\boldsymbol{x}; \boldsymbol{\theta}, \boldsymbol{\phi}, \boldsymbol{s}^2, \mathcal{B})$, where $\boldsymbol{s}^2 := \{s_l^2\}_{l=1}^L$, $\mathcal{B} := (\boldsymbol{B}_1, \cdots, \boldsymbol{B}_L)$ and

$$\mathcal{J}_{\text{SQ-VAE-2}}(\boldsymbol{x}; \boldsymbol{\theta}, \boldsymbol{\phi}, \boldsymbol{s}^2, \mathcal{B}) = \mathbb{E}_{\mathcal{Q}(\boldsymbol{Z}_{1:L}, \tilde{\boldsymbol{Z}}_{1:L} | \boldsymbol{x})} \left[ -\log p_{\boldsymbol{\theta}}(\boldsymbol{x} | \boldsymbol{Z}_{1:L}) + \log \frac{\mathcal{Q}(\boldsymbol{Z}_{1:L}, \tilde{\boldsymbol{Z}}_{1:L} | \boldsymbol{x})}{\mathcal{P}(\boldsymbol{Z}_{1:L}, \tilde{\boldsymbol{Z}}_{1:L})} \right] \qquad (6)$$

Hereafter, we omit the arguments of objective functions for simplicity. By decomposing $\mathcal{Q}$ and $\mathcal{P}$ and substituting parameterizations for the probabilistic parts, we have

$$\mathcal{J}_{\text{SQ-VAE-2}} = \frac{D}{2} \log \sigma^2 + \mathbb{E}_{\mathcal{Q}(\boldsymbol{Z}_{1:L}, \tilde{\boldsymbol{z}}_{1:L} | \boldsymbol{x})} \left[ \frac{\|\boldsymbol{x} - f_{\boldsymbol{\theta}}(\boldsymbol{Z}_{1:L})\|_2^2}{2\sigma^2} + \sum_{l=1}^L \left( \frac{\|\tilde{\boldsymbol{Z}}_l - \boldsymbol{Z}_l\|_F^2}{2s_l^2} - H(\hat{P}_{s_l^2}(\boldsymbol{Z}_l | \tilde{\boldsymbol{Z}}_l)) \right) \right], \quad (7)$$

where $H(\cdot)$ indicates the entropy of a probability mass function and constant terms are omitted. The derivation of Equation (7) is given in Appendix A. The objective function (7) consists of the reconstruction term and the regularization terms for $\boldsymbol{Z}_{1:L}$ and $\tilde{\boldsymbol{Z}}_{1:L}$. The expectation w.r.t. the probability mass function $\hat{P}_{s_l^2}(\boldsymbol{z}_{l,i} = \boldsymbol{b}_k | \tilde{\boldsymbol{Z}}_l)$ can be approximated with the corresponding Gumbel-softmax distribution (Maddison et al., 2017; Jang et al., 2017) in a reparameterizable manner.

---

[1] We empirically found setting $\tilde{\boldsymbol{Z}}_l$ to $\hat{\boldsymbol{Z}}_l$ instead of sampling $\tilde{\boldsymbol{Z}}_l$ from $p_{\boldsymbol{s}^2}(\tilde{z}_{l,i} | \hat{\boldsymbol{Z}}_l)$ leads to better performance (as reported in Takida et al. (2022b); therefore, we follow the procedure in practice.

### 4.2.2 SQ-VAE-2 vs. VQ-VAE-2

The architecture of VQ-VAE-2 is composed in a similar fashion to that of SQ-VAE-2 but is trained by the following objective function:

$$\mathcal{L}_{\text{VQ-VAE-2}} = \|\boldsymbol{x} - \boldsymbol{f_\theta}(\boldsymbol{Z}_{1:L})\|_2^2 + \beta \sum_{l=1}^{L} \|\boldsymbol{G}_\phi^l(\boldsymbol{x}, \boldsymbol{Z}_{1:l-1}) - \text{sg}[\boldsymbol{Z}_l]\|_F^2, \tag{8}$$

where the codebooks are updated with the EMA update in the same manner as the original VQ-VAE. The objective function (8), except for the stop gradient operator and EMA update, can be obtained by setting both $s_l^2$ and $\sigma^2$ to infinity while keeping the ratio of the variances as $s_l^2 = \beta^{-1}\sigma^2$ for $l \in [L]$ in Equation (7). In contrast, since all the parameters but $D$ and $L$ in Equation (7) are optimized, the weight of each term is automatically adjusted during training. Furthermore, SQ-VAE-2 is expected to benefit from the *self-annealing* effect as in the original SQ-VAE (see Section 5.3).

### 4.3 Residual top-down layer

In this subsection, we set $R = 1$ for the simplicity of the demonstration purpose (general case of $R$ is in Appendix B). This means the *bottom-up* and *top-down* paths are connected only at the top layer. We design a *residual top-down* layer for the approximated posterior as in Figure 1d. This layer is to better approximate the target feature with additional assignments of code vectors. By stacking this procedure $L$ times, the feature is approximated as

$$\boldsymbol{H}_\phi(\boldsymbol{x}) \approx \sum_{l=1}^{L} \boldsymbol{Z}_l. \tag{9}$$

Therefore, in this layer, only the information from the higher layers but from the *bottom-up* path is fed to the layer. It is desired that $\sum_{l'=1}^{l+1} \boldsymbol{Z}_{l'}$ approximate the feature better than $\sum_{l'=1}^{l} \boldsymbol{Z}_{l'}$. On this basis, we let the following residual pass through to the next layer:

$$\boldsymbol{G}_\phi^l(\boldsymbol{x}, \boldsymbol{Z}_{1:l-1}) = \boldsymbol{H}_\phi(\boldsymbol{x}) - \sum_{l=1}^{l-1} \boldsymbol{Z}_{l'}. \tag{10}$$

### 4.3.1 RSQ-VAE

We especially instantiate the HQ-VAE only with the *residual top-down* layers in addition to the first layer, which reduces to RSQ-VAE. At this point, by following Equation (6) and omitting constant terms, we can derive the same form of the ELBO objective as Equation (7):

$$\mathcal{J}_{\text{RSQ-VAE}}^{\text{naïve}} = \frac{D}{2} \log \sigma^2 + \mathbb{E}_{\mathcal{Q}(\boldsymbol{Z}_{1:L}, \tilde{\boldsymbol{z}}_{1:L}|\boldsymbol{x})} \left[ \frac{\|\boldsymbol{x} - \boldsymbol{f_\theta}(\boldsymbol{Z}_{1:L})\|_2^2}{2\sigma^2} + \sum_{l=1}^{L} \left( \frac{\|\tilde{\boldsymbol{Z}}_l - \boldsymbol{Z}_l\|_F^2}{2s_l^2} - H(\hat{P}_{s_l^2}(\boldsymbol{Z}_l|\tilde{\boldsymbol{Z}}_l)) \right) \right], \tag{11}$$

where the numerator of the third term corresponds to the evaluation of the residuals $\boldsymbol{H}_\phi(\boldsymbol{x}) - \sum_{l'=1}^{l} \boldsymbol{Z}_{l'}$ for all $l \in [L]$ with the dequantization process. However, we empirically found training the model with the ELBO objective was often unstable. We suspect this is because the objective regularizes $\boldsymbol{Z}_{1:L}$ to make $\sum_{l'=1}^{l} \boldsymbol{Z}_{l'}$ close to the feature for all $l \in [L]$. We hypothesize that this regularization is too strong to regularize the latent representation. To address the issue, we consider conditional distributions not on $(\boldsymbol{Z}_{1:L}, \tilde{\boldsymbol{Z}}_{1:L})$ but on $(\boldsymbol{Z}_{1:L}, \tilde{\boldsymbol{Z}})$, where $\tilde{\boldsymbol{Z}} = \sum_{l=1}^{L} \tilde{\boldsymbol{Z}}_l$. From the reproductive property of Gaussian distribution, the continuous latent variable converted from $\boldsymbol{Z}$ via the stochastic dequantization processes, $\tilde{\boldsymbol{Z}} = \sum_{l=1}^{L} \tilde{\boldsymbol{Z}}_l$, follows the following Gaussian distribution:

$$p_{\boldsymbol{s}^2}(\tilde{\boldsymbol{z}}_i|\boldsymbol{Z}) = \mathcal{N}\left( \tilde{\boldsymbol{z}}_i; \sum_{l=1}^{L} \boldsymbol{z}_{l,i}, \left( \sum_{l=1}^{L} s_l^2 \right) \boldsymbol{I} \right). \tag{12}$$

Table 1: Evaluation on ImageNet (256×256) and FFHQ (1024×1024). RMSE ($\times 10^2$), LPIPS, and SSIM are evaluated using test set. Following Razavi et al. (2019), codebook capacity for discrete latent space is set to $(d_l, K_l) = (32^2, 512), (64^2, 512)$ and $(d_l, K_l) = (32^2, 512), (64^2, 512), (128^2, 512)$ for ImageNet and FFHQ, respectively. We also show codebook perplexity at each layer.

| Dataset | Model | Reconstruction | | | Codebook perplexity | | |
|---------|-------|-------|-------|-------|-------|-------|-------|
| | | RMSE ↓ | LPIPS ↓ | SSIM ↑ | $\exp(H(Q(\boldsymbol{Z}_1)))$ | $\exp(H(Q(\boldsymbol{Z}_2)))$ | $\exp(H(Q(\boldsymbol{Z}_3)))$ |
| ImageNet | VQ-VAE-2 | $6.071 \pm 0.006$ | $0.265 \pm 0.012$ | $0.751 \pm 0.000$ | $106.8 \pm 0.8$ | $288.8 \pm 1.4$ | |
| | SQ-VAE-2 | $4.603 \pm 0.006$ | $0.096 \pm 0.000$ | $0.855 \pm 0.006$ | $406.2 \pm 0.9$ | $355.5 \pm 1.7$ | |
| FFHQ | VQ-VAE-2 | $4.866 \pm 0.291$ | $0.323 \pm 0.012$ | $0.814 \pm 0.003$ | $24.6 \pm 10.7$ | $41.3 \pm 14.0$ | $310.1 \pm 29.6$ |
| | SQ-VAE-2 | $2.118 \pm 0.013$ | $0.166 \pm 0.002$ | $0.909 \pm 0.001$ | $125.8 \pm 9.0$ | $398.7 \pm 14.1$ | $441.3 \pm 7.9$ |

We instead use the following prior distribution to derive the ELBO objective:

$$\mathcal{P}(\boldsymbol{Z}_{1:L}, \tilde{\boldsymbol{Z}}) = \prod_{i=1}^{d_z} \left( \prod_{l=1}^{L} P(\boldsymbol{z}_{l,i}) \right) p_{\boldsymbol{s}^2}(\tilde{\boldsymbol{z}}_i | \boldsymbol{Z}). \tag{13}$$

We now derive the ELBO using the newly established prior and posterior starting from

$$\log p_{\boldsymbol{\theta}}(\boldsymbol{x}) \geq -\mathcal{J}_{\text{RSQ-VAE}}$$
$$= -\mathbb{E}_{\mathcal{Q}(\boldsymbol{Z}_{1:L}, \tilde{\boldsymbol{z}} | \boldsymbol{x})} \left[ -\log p_{\boldsymbol{\theta}}(\boldsymbol{x} | \boldsymbol{Z}_{1:L}) + \log \frac{\mathcal{Q}(\boldsymbol{Z}_{1:L}, \tilde{\boldsymbol{Z}} | \boldsymbol{x})}{\mathcal{P}(\boldsymbol{Z}_{1:L}, \tilde{\boldsymbol{Z}})} \right]. \tag{14}$$

The above objective is further simplified as

$$\mathcal{J}_{\text{RSQ-VAE}} = \frac{D}{2} \log \sigma^2 + \mathbb{E}_{\mathcal{Q}(\boldsymbol{Z}_{1:L}, \tilde{\boldsymbol{z}}_{1:L} | \boldsymbol{x})} \left[ \frac{\|\boldsymbol{x} - \boldsymbol{f}_{\boldsymbol{\theta}}(\boldsymbol{Z}_{1:L})\|_2^2}{2\sigma^2} + \frac{\|\tilde{\boldsymbol{Z}} - \boldsymbol{Z}\|_F^2}{2\sum_{l=1}^{L} s_l^2} - \sum_{l=1}^{L} H(\hat{P}_{s_l^2}(\boldsymbol{Z}_l | \tilde{\boldsymbol{Z}}_l)) \right] \tag{15}$$

where the third term is different from that in Equation (11) and its numerator evaluates only the overall quantization error $\boldsymbol{H}_{\boldsymbol{\phi}}(\boldsymbol{x}) - \sum_{l=1}^{L} \boldsymbol{Z}_l$ with the dequantization process.

### 4.3.2 RSQ-VAE vs. RQ-VAE

RQ-VAE and RSQ-VAE both learn discrete representation in a coarse-to-fine manner, but RQ-VAE adopts a deterministic RQ scheme to achieve Equation (9), where RQ-VAE is trained with the following objective function:

$$\mathcal{L}_{\text{RQ-VAE}} = \|\boldsymbol{x} - \boldsymbol{f}_{\boldsymbol{\theta}}(\boldsymbol{Z}_{1:L})\|_2^2 + \beta \sum_{l=1}^{L} \left\| \boldsymbol{H}_{\boldsymbol{\phi}}(\boldsymbol{x}) - \text{sg} \left[ \sum_{l'=1}^{l} \boldsymbol{Z}_{l'} \right] \right\|_F^2, \tag{16}$$

where the codebooks are updated with the EMA update in the same manner as VQ-VAE. The second term of Equation (16) resembles the third term of Equation (11), which strongly enforces certain degree of reconstruction even only with partial information from the higher layers. RQ-VAE is beneficial from such a regularization term, which leads to stable training. However, in RSQ-VAE, this regularization deteriorates the reconstruction performance. Instead, we use Equation (15) as the objective, which regularizes the latent representation by taking into account only accumulated information from all layers.

**Remark.** HQ-VAE has favorable properties similar to SQ-VAE. The training scheme does not require hyperparameters except for a temperature parameter of Gumbel-softmax approximation (see Equations (7) and (15)). Furthermore, the derived models can benefit from the *self-annealing effect* as in SQ-VAE, which is empirically shown in Section 5.3.

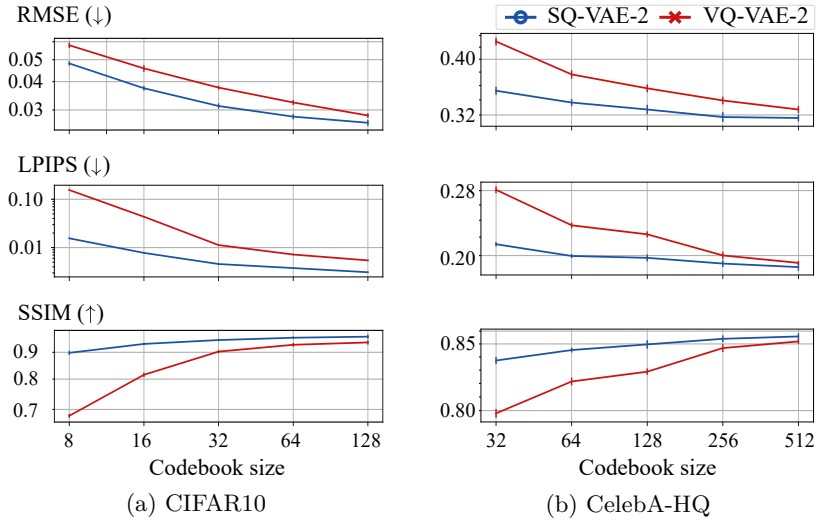

(a) CIFAR10        (b) CelebA-HQ

Figure 2: Impact of codebook capacity on reconstruction is investigated on (a) CIFAR10 and (b) CelebA-HQ. Two and three layers are tested on CIFAR10 and CelebA-HQ, respectively.

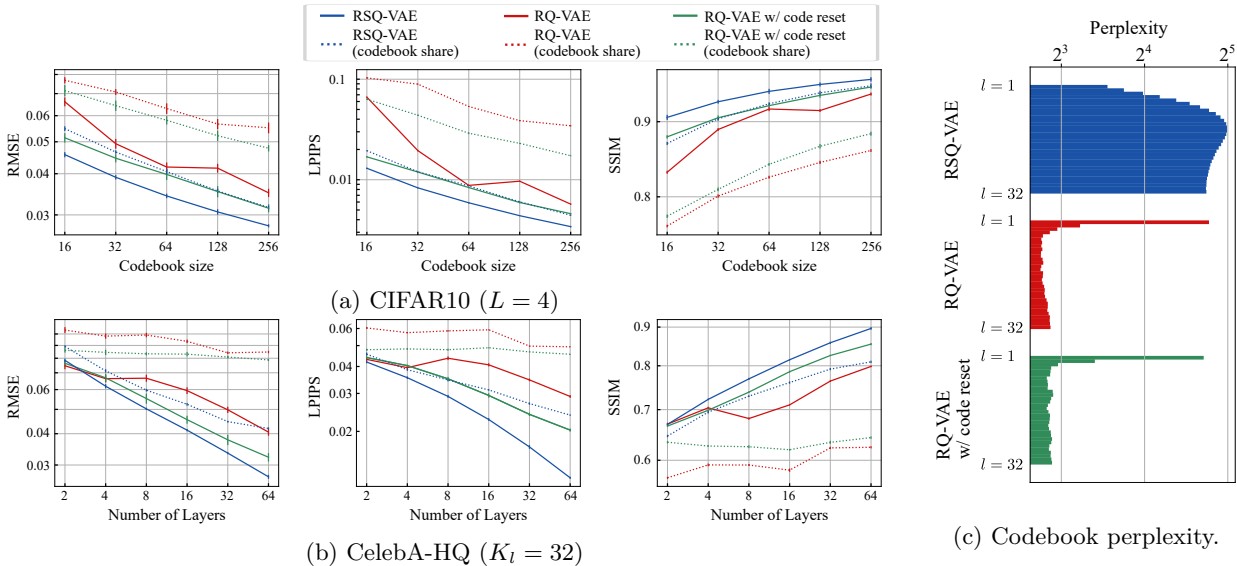

Figure 3: Impact of codebook capacity on reconstruction is investigated on (a) CIFAR10 and (b) CelebA-HQ. (c) Codebook perplexity at each layer is plotted, wheremodels with 32 layers are trained on CelebA-HQ and all layers share same codebook.

# 5 Experiments

We comprehensively examine SQ-VAE-2 and RSQ-VAE and visualize the effects of their individual *top-down* paths. In Secs. 5.1 and 5.2, we comprehensively compare SQ-VAE-2 and RSQ-VAE with VQ-VAE-2 and RQ-VAE, respectively to show our framework improves reconstruction performance against the baselines. We basically conduct the comparison with various latent capacities to evaluate our methods in a rate–distortion (RD) sense (Alemi et al., 2017). In addition, we show that RSQ-VAE trained with a perceptual loss (Johnson et al., 2016) is competitive with the state-of-the-art model based on RQ-VAE. Furthermore, we test HQ-VAE on an audio dataset to show that it is applicable to a different modality. In Section 5.3, we investigate the characteristics of the *injected top-down* and *residual top-down* layers with visualization (Section 5.3). Unless otherwise noted, we use the same network architecture in all models and set the codebook dimension to $d_b = 64$. The experimental details are given in Appendix C.

## 5.1 SQ-VAE-2 vs. VQ-VAE-2

We compare our SQ-VAE-2 with VQ-VAE-2 from the aspects of reconstruction accuracy and codebook utilization. We first investigate their performance on CIFAR10 (Krizhevsky et al., 2009) and CelebA-HQ (256×256) in various codebook settings: the configurations for the hierarchical structure and numbers of code vectors ($K_l$). We evaluate the reconstruction accuracy in terms of a Euclidean metric and two perceptual metrics: the root mean squared error (RMSE), structure similarity index (SSIM) (Wang et al., 2004), and learned perceptual image patch similarity (LPIPS) (Zhang et al., 2018). As shown in Figure 2, SQ-VAE-2 achieves better reconstruction accuracy in all cases. The difference of the performance between the two models is noticeable when the codebook size is small.

**Comparison on large-scale datasets.** Next, we demonstrate that SQ-VAE-2 outperforms VQ-VAE-2 on ImageNet (256×256) (Deng et al., 2009) and FFHQ (1024×1024) (Karras et al., 2019) in the same latent settings as in Razavi et al. (2019). As shown in Table 1, SQ-VAE-2 achieves better reconstruction performance in terms of RMSE, LPIPS, and SSIM than VQ-VAE-2, which is a similar tendency as in the comparison on CIFAR10 and CelebA-HQ. Furthermore, we measure codebook utilization per layer by using the perplexity of latent variables. The codebook perplexity is defined as $\exp(H(Q(\boldsymbol{Z}_l)))$, where $Q(\boldsymbol{Z}_l)$ is a marginalized distribution of Equation (4) with $\boldsymbol{x} \sim p_d(\boldsymbol{x})$. The perplexity ranges from 1 to the number of code vectors ($K_l$) by its definition. SQ-VAE-2 achieves higher codebook perplexities than VQ-VAE-2 at all the layers, whereas the higher layers are not effectively used in VQ-VAE-2. In particular, the perplexity values at the top layer in the case of VQ-VAE-2 is extremely low, which is the sign of layer collapse.

## 5.2 RSQ-VAE vs. RQ-VAE

We compare our RSQ-VAE with RQ-VAE using the same metrics as in Section 5.1. As codebook reset is used in the original study of RQ-VAE (Zeghidour et al., 2021; Lee et al., 2022a) to prevent codebook collapse, we add RQ-VAE with this technique to the baselines. We do not apply it to RSQ-VAE because it is not explainable in the variational Bayes framework. In addition, Lee et al. (2022a) proposed to share codebook for all the layers, i.e., $\boldsymbol{B}_l = \boldsymbol{B}$ for $l \in [L]$ to enhance the utility of the codes. We test both RSQ-VAE and RQ-VAE with and without the codebook share. We first investigate their performances on CIFAR10 and CelebA-HQ (256×256) in various settings: the number of times of quantization step ($l$) and number of code vectors ($K_l$). As shown in Figures 3a and 3b, RSQ-VAE achieves better reconstruction accuracy in terms of RMSE, SSIM, and LPIPS than the baselines although codebook reset overall enhances the performance of RQ-VAE. When the codebook is shared for all layers, the performance difference is remarkable, with a noticeable difference in how codes are used. Interestingly, more codes are assigned in the bottom layers in RSQ-VAE, unlike RQ-VAEs, as shown in Figure 3c. RSQ-VAE captures the coarse information with a relatively small number of codes, and refines the reconstruction with larger bits at the bottom layers.

**Improvement in perceptual quality.** Next, we use the same network architecture as that used in Lee et al. (2022a) and set the codebook dimension to $d_b = 256$ for fair comparison with their RQ-VAE. We train RSQ-VAE on FFHQ with an LPIPS loss (Zhang et al., 2018) (see Appendix C.4) and compare it with their

Table 2: Evaluation on UrbanSound8K. RMSE is evaluated using test set. Network architecture follows Liu et al. (2021). Codebook size is set to $K_l = 8$.

| Model | Number of Layers | RMSE ↓ |
|---|---|---|
| RQ-VAE | 4 | $0.506 \pm 0.018$ |
| | 8 | $0.497 \pm 0.057$ |
| RSQ-VAE | 4 | $0.427 \pm 0.014$ |
| | 8 | $0.314 \pm 0.013$ |

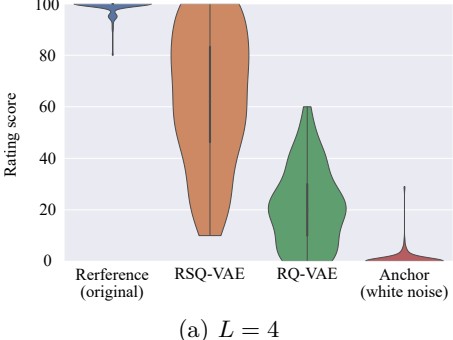
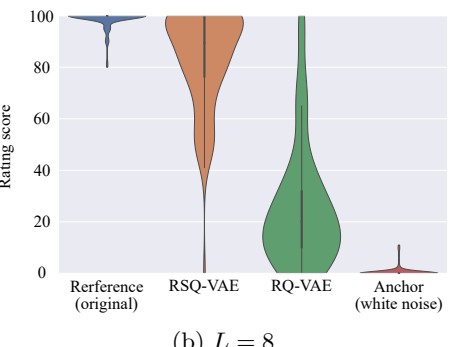

(a) $L = 4$    (b) $L = 8$

Figure 4: Violin plots of MUSHRA listening test results on UrbanSound8K test set in the cases of (a) 4 layers and (b) 8 layers. The white dots indicate the median scores, and the tops and bottoms of thick vertical lines indicate the first and third quartiles, respectively.

RQ-VAE in terms of Fréchet Inception Distance (FID) (Heusel et al., 2017). The reconstructed FID (rFID) of RSQ-VAE is 8.47, whereas that of their RQ-VAE is 7.29. Note that we do not use an adversarial loss for training RSQ-VAE but their RQ-VAE was trained with the combination of an LPIPS loss and adversarial loss. This means that our RSQ-VAE achieves competitive performance without an adversarial loss. We leave combining an adversarial loss with HQ-VAE for future work.

**Validation on an audio dataset.** To validate the effectiveness of RSQ-VAE in the audio domain, we compare it with RQ-VAE by the reconstruction of the normalized log-Mel spectrogram using an environmental sound dataset: UrbanSound8K (Salamon et al., 2014). We follow the same network architecture used in an audio generation paper (Liu et al., 2021), which deploys multi-scale convolutional layers with varied kernel sizes to capture the local and global features of audio signals in the time-frequency domain (Xian et al., 2021). Codebook size is set to $K_l = 8$. Number of layers is set to 4 and 8, and all the layers share the same codebook. We run each trial with five different random seeds and obtain the average and standard deviation of RMSEs. As shown in Table 2, RSQ-VAE achieves better average RMSEs than RQ-VAE across different numbers of layers on the audio dataset. To evaluate the perceptual quality of our results, we also perform a subjective listening test using the multiple stimulus hidden reference anchor (MUSHRA) protocol (Series, 2014) on an audio web evaluation tool (Schoeffler et al., 2018). We randomly select an audio signal from the UrbanSound8K test set for the practicing part. In the test part, we extract ten samples by randomly selecting an audio signal per class from the test set. We prepare four samples for each signal: reconstructed samples from RQ-VAE and RSQ-VAE, a white noise signal as a hidden anchor, and an original sample as a hidden reference. Because RQ-VAE and RSQ-VAE are applied for the normalized log-Mel spectrograms, we use a vocoder in the audio generation paper (Liu et al., 2021; Kong et al., 2020) to convert the reconstructed spectrograms to the waveform samples. As the upper bound quality of the reconstructed waveform is limited to the vocoder result of the original spectrogram, we use the vocoder result for the reference. After listening to the reference, assessors are asked to rate 0 to 100 the four different samples according to similarity relative to the reference. After a post-screening of assessors (Series, 2014), there are a total of 10 assessors in the test. We show the violin plot results of the listening test in Figure 4. The figure shows that RSQ-VAE achieves better median listening scores than RQ-VAE. Especially when the number of layers is 8, the plots of RSQ-VAE show better scores with a large margin from the those of RQ-VAE.

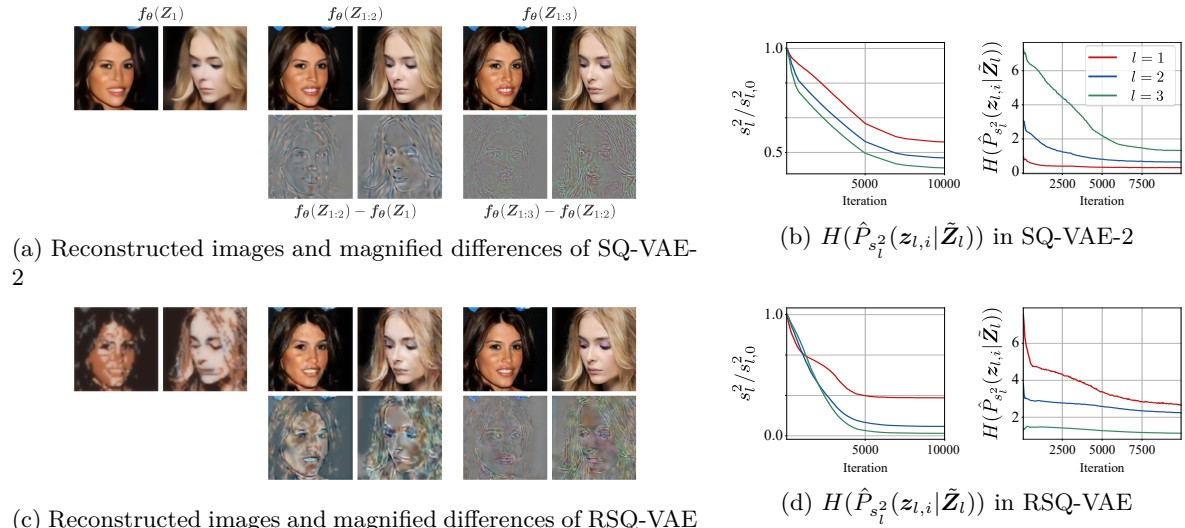

(a) Reconstructed images and magnified differences of SQ-VAE-2

(b) $H(\hat{P}_{s_l^2}(z_{l,i}|\tilde{Z}_l))$ in SQ-VAE-2

(c) Reconstructed images and magnified differences of RSQ-VAE

(d) $H(\hat{P}_{s_l^2}(z_{l,i}|\tilde{Z}_l))$ in RSQ-VAE

Figure 5: Reconstructed samples with partial layers in (a) SQ-VAE-2 and (c) RSQ-VAE. Top row shows reconstructed images while bottom one shows added components at each layer. For $l = 1, 2, 3$ latent capacity is set to $(d_l, K_l) = (16^2, 256), (32^2, 16), (64^2, 4)$ and $(d_l, K_l) = (32^2, 4), (32^2, 16), (32^2, 256)$, respectively. Notice that the numbers of bits of these models are equal at each layer. For reasonable visualization, we apply *progressive coding* to SQ-VAE-2, which induces progressive compression (see Appendix C.5). (b) and (d) We plot variance parameter $s_l^2$ normalized by initial value $s_{l,0}^2$ and average entropy of quantization process $(H(\hat{P}_{s_l^2}(z_{l,i}|\tilde{Z}_l)))$ at each layer.

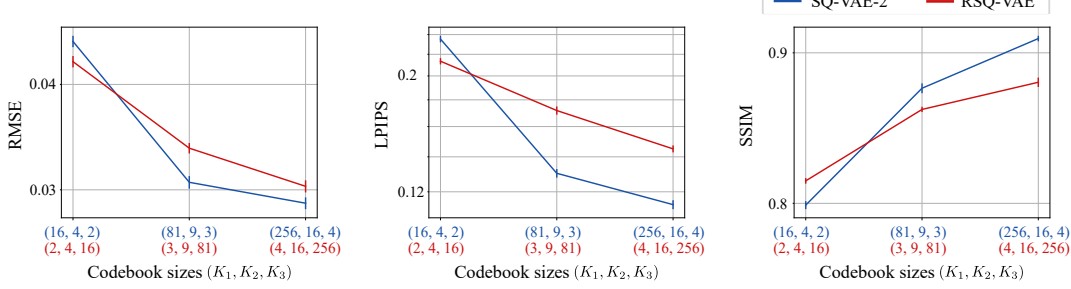

Figure 6: Comparison of SQ-VAE-2 and RSQ-VAE under three latent capacity cases. As for x-axes, we put $(K_1, K_2, K_3)$ for SQ-VAE-2 and RSQ-VAE in blue and red, respectively. Note that $(d_1, d_2, d_3) = (16, 32, 64)$ and $(d_1, d_2, d_3) = (32, 32, 32)$ for SQ-VAE-2 and RSQ-VAE as in Figure 5, and each x-axis value indicate the same latent capacity. SQ-VAE-2 outperforms RSQ-VAE in relatively large latent capacity settings. In contrast, RSQ-VAE achieves better reconstruction performance for the case of higher compression rate.

## 5.3 Empirical study of top-down layers

In this section, we focus on visualizing the obtained discrete representations instead of comparing their reconstruction performance. This will provide insights into the characteristics of the *top-down* layers. We train both, SQ-VAE-2 and RSQ-VAE, with three layers on CelebA-HQ (Karras et al., 2018), respectively. Figure 5 shows the progressively reconstructed images. For demonstration purpose, we incorporate *progressive coding* (Shu & Ermon, 2022) to SQ-VAE-2 to make the reconstructed images only with the top layers interpretable. We note that *progressive coding* is not applied other than the illustration of Figure 5. Both, SQ-VAE-2 and RSQ-VAE, share the similarity that the higher layers generate the coarse part of the image while the lower layers complement them with details. However, comparing the two, we observe that in SQ-VAE-2, the additionally generated components (bottom row in Figure 5a) in each layer have different resolutions. We conjecture that the different layer-dependent resolutions $\boldsymbol{H}_\phi^{r(l)}(\boldsymbol{x})$, which are injected into

the *top-down* layers, contain different information. This implies that we may obtain more interpretable discrete representations if we can explicitly manipulate the extracted features in the *bottom-up* path to provide $\boldsymbol{H}_{\phi}^{r^{(l)}}(\boldsymbol{x})$ giving them more semantic meaning (e.g., texture or color). In contrast, RSQ-VAE seems to obtain a different discrete representation which resembles more a decomposition. This might be due to its approximated expansion in Equation (9). Moreover, we can observe from Figures 5b and 5d that the *top-down* layers also benefit from the *self-annealing* effect.

In Appendix C.3, we explore combining the two layers to form a hybrid model. We observe that individual layers in a hybrid model produce similar effects as if they would be used alone. That is, outputs from *injected top-down* layers have better resolution and *residual top-down* layers refine upon certain decomposition. Since these two layers enjoy distinct refining mechanisms, a hybrid model may bring a more flexible approximation to the posterior distribution.

Lastly, we compare reconstruction performances of SQ-VAE-2 and RSQ-VAE under the same architectures for the *bottom-up* and *top-down* paths. We follow the same experimental condition as that for visualization of Figure 5. We compare them in three different compression rates by changing the number of code vectors $K_l$. Interestingly, we can see in Figure 6 that SQ-VAE-2 achieve better reconstruction performance in the case of the lower compression rate, whereas RSQ-VAE reconstructs the original images better than SQ-VAE-2 in the case of higher compression rate.

## 6 Discussion

### 6.1 Conclusion

We propose HQ-VAE, a general VAE approach that learns hierarchical discrete representations. HQ-VAE is formulated within the variational Bayes framework as a stochastic quantization technique, which (1) greatly reduces the number of hyperparameters to be tuned (only the one from the Gumbel-softmax trick), and (2) enhances codebook usage without any heuristics thanks to the *self-annealing* effect. We instantiate the general HQ-VAE with two types of posterior approximators for the discrete latent representations, which lead to SQ-VAE-2 and RSQ-VAE. These two novel variants share a similar design of information passing as VQ-VAE-2 and RQ-VAE, respectively, but their latent representations are quantized stochastically. Our experiments show that SQ-VAE-2 and RSQ-VAE outperform their individual baselines with better reconstruction and more efficient codebook usages in the image as well as audio domain.

### 6.2 Concluding remarks

First, VQ-VAE-2 and RQ-VAE can be basically replaced with SQ-VAE-2 and RSQ-VAE for improvement in many previous work in terms of reconstruction accuracy and efficient codebook usage. SQ-VAE-2 and RSQ-VAE achieve better RD curve than the baselines, which means, HQ-VAEs achieve (i) better reconstruction performance under the same latent capacities, and (ii) comparable reconstruction performance with the higher compression rate, compared with the baselines. Furthermore, our approach eliminates the need for the repetition of tuning many hyper-parameters and the introduction of ad-hoc techniques. Both SQ-VAE-2 and RSQ-VAE are applicable to generative modeling with the additional training of prior models as in a bunch of previous work. Generally, compression models with better RD curves are more feasible for the prior models (Rombach et al., 2022), hence the replacement of VQ-VAE-2 and RQ-VAE with SQ-VAE-2 and RSQ-VAE is beneficial even for generative modeling. Recently, RQ-VAE is more often used for the generation tasks than VQ-VAE-2 due to the severe instability issue of VQ-VAE-2, i.e., layer collapse (Dhariwal et al., 2020). However, we believe that the proposal of SQ-VAE-2 has a potential to advocate the use of such a hierarchical model for generation tasks since it mitigates the issue greatly.

SQ-VAE-2 and RSQ-VAE have their own unique advantages as follows. SQ-VAE-2 was shown to be able to learn multi-resolution discrete representation thanks to the design of *bottom-up* path with the pooling operators (see Figure 5a). The results implied that further semantic disentanglement of the discrete representation might be possible by adopting specific inductive architectural components in the *bottom-up* path, which is an interesting future direction. As a second note, in the case of lower compression rates, SQ-VAE-2

outperforms RSQ-VAE in reconstruction performance (see Section 5.3). It hints at that SQ-VAE-2 might be appropriate especially for high-fidelity generation tasks when the prior model could be larger as in Razavi et al. (2019). In contrast, one of the strengths of RSQ-VAE is that one can easily accommodate different compression rates only by changing the number of layers during the inference (without changing or retraining the model). Furthermore, in the case of higher compression rates, RSQ-VAE outperforms SQ-VAE-2 in reconstruction performance (see Section 5.3). The properties are suitable for the application to neural codec as in Zeghidour et al. (2021); Défossez et al. (2022).

### 6.3 Future work

As future work, we will incorporate adversarial training into HQ-VAE which is expected to further enhance the perceptual quality of the reconstructed data. As discussed in Section 5.3, we will also explore the feasibility of explicitly manipulating the injected information into the *top-down* layers to obtain discrete representations with semantic meaning. At last, we explore one of the applications to image generation by training a prior on extracted discrete representations with HQ-VAE in Section D. Nevertheless, this work is focusing on providing a unified variational Bayesian framework of hierarchical quantization. Its downstream applications such as content generation or neural codec will also be considered as future work.

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

## A Derivations

### A.1 SQ-VAE-2

The ELBO of SQ-VAE-2 is formulated by using Bayes' theorem as

$$\log p_{\boldsymbol{\theta}}(\boldsymbol{x}) \geq \log p_{\boldsymbol{\theta}}(\boldsymbol{x}) - D_{\mathrm{KL}}(\mathcal{Q}(\boldsymbol{Z}_{1:L}, \tilde{\boldsymbol{Z}}_{1:L}|\boldsymbol{x}) \| \mathcal{P}(\boldsymbol{Z}_{1:L}, \tilde{\boldsymbol{Z}}_{1:L}|\boldsymbol{x}))$$

$$= \mathbb{E}_{\mathcal{Q}(\boldsymbol{Z}_{1:L}, \tilde{\boldsymbol{Z}}_{1:L}|\boldsymbol{x})} \left[ \log \frac{p_{\boldsymbol{\theta}}(\boldsymbol{x})\mathcal{P}(\boldsymbol{Z}_{1:L}, \tilde{\boldsymbol{Z}}_{1:L}|\boldsymbol{x})}{\mathcal{Q}(\boldsymbol{Z}_{1:L}, \tilde{\boldsymbol{Z}}_{1:L}|\boldsymbol{x})} \right]$$

$$= \mathbb{E}_{\mathcal{Q}(\boldsymbol{Z}_{1:L}, \tilde{\boldsymbol{Z}}_{1:L}|\boldsymbol{x})} \left[ \log p_{\boldsymbol{\theta}}(\boldsymbol{x}|\boldsymbol{Z}_{1:L}) - \log \frac{\mathcal{Q}(\boldsymbol{Z}_{1:L}, \tilde{\boldsymbol{Z}}_{1:L}|\boldsymbol{x})}{\mathcal{P}(\boldsymbol{Z}_{1:L}, \tilde{\boldsymbol{Z}}_{1:L})} \right]$$

$$= \mathbb{E}_{\mathcal{Q}(\boldsymbol{Z}_{1:L}, \tilde{\boldsymbol{Z}}_{1:L}|\boldsymbol{x})} \left[ \log p_{\boldsymbol{\theta}}(\boldsymbol{x}|\boldsymbol{Z}_{1:L}) - \sum_{l=1}^{L} \sum_{i=1}^{d_l} \left( \log \frac{p_{s_l^2}(\tilde{z}_{l,i}|\hat{\boldsymbol{Z}}_l)}{p_{s_l^2}(\tilde{z}_{l,i}|\boldsymbol{Z}_l)} + \log \frac{\hat{P}_{s_l^2}(\boldsymbol{z}_{l,i}|\tilde{\boldsymbol{Z}}_l)}{P(\boldsymbol{z}_{l,i})} \right) \right]$$

$$= \mathbb{E}_{\mathcal{Q}(\boldsymbol{Z}_{1:L}, \tilde{\boldsymbol{Z}}_{1:L}|\boldsymbol{x})} \left[ \log p_{\boldsymbol{\theta}}(\boldsymbol{x}|\boldsymbol{Z}_{1:L}) + \sum_{l=1}^{L} \sum_{i=1}^{d_l} \left( \log \frac{p_{s_l^2}(\tilde{z}_{l,i}|\boldsymbol{Z}_l)}{p_{s_l^2}(\tilde{z}_{l,i}|\hat{\boldsymbol{Z}}_l)} + H(\hat{P}_{s_l^2}(\boldsymbol{z}_{l,i}|\tilde{\boldsymbol{Z}}_l)) - \log K_l \right) \right]. \quad (17)$$

Since the probabilistic parts are modeled as Gaussian distributions, the first and second terms can be calculated as

$$\log p_{\boldsymbol{\theta}}(\boldsymbol{x}|\boldsymbol{Z}_{1:L}) = \log \mathcal{N}(\boldsymbol{x}; \boldsymbol{f}_{\boldsymbol{\theta}}(\boldsymbol{Z}_{1:L}), \sigma^2 \boldsymbol{I})$$

$$= -\frac{D}{2} \log(2\pi\sigma^2) - \frac{1}{2\sigma^2} \|\boldsymbol{x} - \boldsymbol{f}_{\boldsymbol{\theta}}(\boldsymbol{x})\|_2^2 \quad \text{and} \quad (18)$$

$$\mathbb{E}_{\mathcal{Q}(\boldsymbol{Z}_{1:L}, \tilde{\boldsymbol{Z}}_{1:L}|\boldsymbol{x})} \left[ \frac{p_{s_l^2}(\tilde{z}_{l,i}|\boldsymbol{Z}_l)}{p_{s_l^2}(\tilde{z}_{l,i}|\hat{\boldsymbol{Z}}_l)} \right] = \mathbb{E}_{\mathcal{Q}(\boldsymbol{Z}_{1:L}, \tilde{\boldsymbol{Z}}_{1:L}|\boldsymbol{x})} \left[ -\frac{1}{2s_l^2} \|\tilde{z}_{l,i} - z_{l,i}\|_2^2 + \frac{1}{2s_l^2} \|\tilde{z}_{l,i} - \hat{z}_{l,i}\|_2^2 \right]$$

$$= -\mathbb{E}_{\mathcal{Q}(\boldsymbol{Z}_{1:L}, \tilde{\boldsymbol{Z}}_{1:L}|\boldsymbol{x})} \left[ \frac{1}{2s_l^2} \|\tilde{z}_{l,i} - z_{l,i}\|_2^2 \right] + \frac{d_b}{2}. \quad (19)$$

By substituting Equations (18) and (19) into Equation (17), we have Equation (7), where we use $\tilde{\boldsymbol{Z}}_l = \hat{\boldsymbol{Z}}_l$ instead of sampling it in practical implementation.

### A.2 RSQ-VAE

The ELBO of RSQ-VAE is formulated by using Bayes' theorem as

$$\log p_{\boldsymbol{\theta}}(\boldsymbol{x}) \geq \log p_{\boldsymbol{\theta}}(\boldsymbol{x}) - D_{\mathrm{KL}}(\mathcal{Q}(\boldsymbol{Z}_{1:L}, \tilde{\boldsymbol{Z}}|\boldsymbol{x}) \| \mathcal{P}(\boldsymbol{Z}_{1:L}, \tilde{\boldsymbol{Z}}|\boldsymbol{x}))$$

$$= \mathbb{E}_{\mathcal{Q}(\boldsymbol{Z}_{1:L}, \tilde{\boldsymbol{Z}}_{1:L}|\boldsymbol{x})} \left[ \log \frac{p_{\boldsymbol{\theta}}(\boldsymbol{x})\mathcal{P}(\boldsymbol{Z}_{1:L}, \tilde{\boldsymbol{Z}}|\boldsymbol{x})}{\mathcal{Q}(\boldsymbol{Z}_{1:L}, \tilde{\boldsymbol{Z}}|\boldsymbol{x})} \right]$$

$$= \mathbb{E}_{\mathcal{Q}(\boldsymbol{Z}_{1:L}, \tilde{\boldsymbol{Z}}|\boldsymbol{x})} \left[ \log p_{\boldsymbol{\theta}}(\boldsymbol{x}|\boldsymbol{Z}_{1:L}) - \log \frac{\mathcal{Q}(\boldsymbol{Z}_{1:L}, \tilde{\boldsymbol{Z}}|\boldsymbol{x})}{\mathcal{P}(\boldsymbol{Z}_{1:L}, \tilde{\boldsymbol{Z}})} \right]$$

$$= \mathbb{E}_{\mathcal{Q}(\boldsymbol{Z}_{1:L}, \tilde{\boldsymbol{Z}}|\boldsymbol{x})} \left[ \log p_{\boldsymbol{\theta}}(\boldsymbol{x}|\boldsymbol{Z}_{1:L}) - \sum_{i=1}^{d_l} \log \frac{p_{\boldsymbol{s}^2}(\tilde{z}_i|\hat{\boldsymbol{Z}})}{p_{\boldsymbol{s}^2}(\tilde{z}_i|\boldsymbol{Z})} - \sum_{l=1}^{L} \sum_{i=1}^{d_l} \log \frac{\hat{P}_{s_l^2}(\boldsymbol{z}_{l,i}|\tilde{\boldsymbol{Z}}_l)}{P(\boldsymbol{z}_{l,i})} \right]$$

$$= \mathbb{E}_{\mathcal{Q}(\boldsymbol{Z}_{1:L}, \tilde{\boldsymbol{Z}}|\boldsymbol{x})} \left[ \log p_{\boldsymbol{\theta}}(\boldsymbol{x}|\boldsymbol{Z}_{1:L}) + \sum_{i=1}^{d_l} \log \frac{p_{\boldsymbol{s}^2}(\tilde{z}_i|\boldsymbol{Z})}{p_{\boldsymbol{s}^2}(\tilde{z}_i|\hat{\boldsymbol{Z}})} + \sum_{l=1}^{L} \sum_{i=1}^{d_l} H(\hat{P}_{s_l^2}(\boldsymbol{z}_{l,i}|\tilde{\boldsymbol{Z}}_l)) - \log K_l \right]. \quad (20)$$

Since the probabilistic parts are modeled as Gaussian distributions, the second term can be calculated as

$$\mathbb{E}_{\mathcal{Q}(\boldsymbol{Z}_{1:L}, \tilde{\boldsymbol{Z}}|\boldsymbol{x})} \left[ \frac{p_{\boldsymbol{s}^2}(\tilde{z}_i|\boldsymbol{Z})}{p_{\boldsymbol{s}^2}(\tilde{z}_i|\hat{\boldsymbol{Z}})} \right] = \mathbb{E}_{\mathcal{Q}(\boldsymbol{Z}_{1:L}, \tilde{\boldsymbol{z}}|\boldsymbol{x})} \left[ -\frac{1}{2\sum_{l=1}^{L} s_l^2} \|\tilde{z}_i - z_i\|_2^2 + \frac{1}{2\sum_{l=1}^{L} s_l^2} \|\tilde{z}_i - \hat{z}_i\|_2^2 \right]$$

$$= -\mathbb{E}_{\mathcal{Q}(\boldsymbol{Z}_{1:L}, \tilde{\boldsymbol{z}}|\boldsymbol{x})} \left[ \frac{1}{2\sum_{l=1}^{L} s_l^2} \|\tilde{z}_i - z_i\|_2^2 \right] + \frac{d_b}{2}. \quad (21)$$

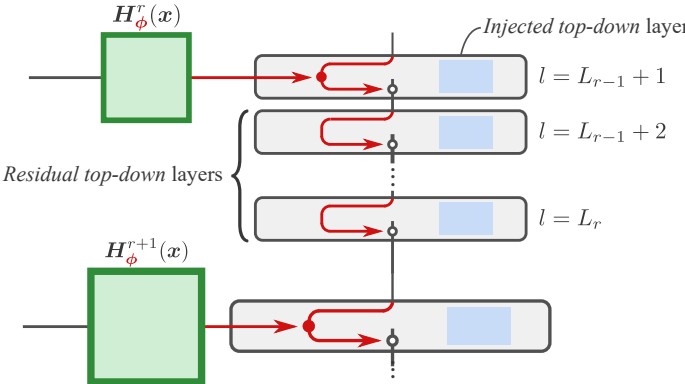

Figure 7: *Top-down* layers corresponding to the *r*th resolution in hybrid model in Appendix B.

By substituting Equations (18) and (21) into (20), we have Equation (15), where we use $\tilde{\boldsymbol{Z}} = \boldsymbol{H}_{\phi}(\boldsymbol{x})$ instead of sampling it in practical implementation. We have derived the ELBO objective in the case of $R = 1$. We extend the model to the general case of $R$, equivalent to the hybrid model, in Appendix B.

## B   Hybrid model

We provide the ELBO of a hybrid model, where the two types of *top-down* layers are combinatorially used to build a *top-down* path as in Figure 7. We introduce some extra notations: $L_r$ indicates the number of layers corresponding to the resolutions from the first to $r$th order; $\ell_r := \{L_{r-1}+1, \cdots, L_r\}$ is a set of all the layers corresponding to the resolution $r$; and the output of the encoding block in the $(L_{r-1}+1)$th layer is denoted as $\tilde{\boldsymbol{G}}_{\phi}^{r}(\boldsymbol{H}_{\phi}^{r}(\boldsymbol{x}), \boldsymbol{Z}_{1:L_{r-1}})$. In Figure 7, the quantized variables $\boldsymbol{Z}_{\ell_r}$ aim at approximating the variable encoded at $l = L_{r-1} + 1$ as

$$\tilde{\boldsymbol{G}}_{\phi}^{r}(\boldsymbol{H}_{\phi}^{r}(\boldsymbol{x}), \boldsymbol{Z}_{1:L_{r-1}}) \approx \sum_{l \in \ell_r} \boldsymbol{Z}_l =: \boldsymbol{Y}_r. \tag{22}$$

On this basis, the $l$th *top-down* layer quantizes the following information:

$$\hat{\boldsymbol{Z}}_l = \boldsymbol{G}_{\phi}^{l}(\boldsymbol{x}, \boldsymbol{Z}_{1:l-1}) = \begin{cases} \boldsymbol{H}_{\phi}^{1}(\boldsymbol{x}) & (l = 1) \\ \tilde{\boldsymbol{G}}_{\phi}^{r(l)}(\boldsymbol{H}_{\phi}^{r(l)}(\boldsymbol{x}), \boldsymbol{Z}_{1:L_{r(l)-1}}) - \sum_{l'=L_{r(l)-1}+1}^{l} \boldsymbol{Z}_{l'} & (l > 1). \end{cases} \tag{23}$$

To derive the ELBO objective, we consider conditional distributions on $(\boldsymbol{Z}_{1:L}, \tilde{\boldsymbol{Y}}_{1:R})$, where $\tilde{\boldsymbol{Y}}_r := \sum_{l \in \ell_r} \tilde{\boldsymbol{Z}}_l$. From the reproductive property of Gaussian distribution, the continuous latent variable converted from $\boldsymbol{Y}_r$ via the stochastic dequantization processes, $\tilde{\boldsymbol{Z}}_{\ell_r}$, follows the following Gaussian distribution:

$$p_{\boldsymbol{s}_r^2}(\tilde{\boldsymbol{y}}_{r,i} | \boldsymbol{Z}_{\ell_r}) = \mathcal{N}\left(\tilde{\boldsymbol{y}}_{r,i}; \sum_{l \in \ell_r} \boldsymbol{z}_{l,i}, \left(\sum_{l \in \ell_r} s_l^2\right) \boldsymbol{I}\right), \tag{24}$$

where $\boldsymbol{s}_r^2 := \{s_l^2\}_{l \in \ell_r}$. We use the following prior distribution to derive the ELBO objective:

$$\mathcal{P}(\boldsymbol{Z}_{1:L}, \tilde{\boldsymbol{Y}}_{1:R}) = \prod_{r=1}^{R} \prod_{i=1}^{d_r} \left(\prod_{l \in \ell_r} P(\boldsymbol{z}_{l,i})\right) p_{\boldsymbol{s}_r^2}(\tilde{\boldsymbol{y}}_{r,i} | \boldsymbol{Z}_{\ell_r}), \tag{25}$$

where $d_r := d_l$ for $l \in \ell_r$. With the prior and posterior distributions, the ELBO of the hybrid model is formulated by using Bayes' theorem as

$$\log p_{\boldsymbol{\theta}}(\boldsymbol{x}) \geq \log p_{\boldsymbol{\theta}}(\boldsymbol{x}) - D_{\text{KL}}(\mathcal{Q}(\boldsymbol{Z}_{1:L}, \tilde{\boldsymbol{Y}}_{1:R}|\boldsymbol{x}) \,\|\, \mathcal{P}(\boldsymbol{Z}_{1:L}, \tilde{\boldsymbol{Y}}_{1:R}|\boldsymbol{x}))$$

$$= \mathbb{E}_{\mathcal{Q}(\boldsymbol{Z}_{1:L}, \tilde{\boldsymbol{Y}}_{1:R}|\boldsymbol{x})} \left[ \log \frac{p_{\boldsymbol{\theta}}(\boldsymbol{x})\mathcal{P}(\boldsymbol{Z}_{1:L}, \tilde{\boldsymbol{Y}}_{1:R}|\boldsymbol{x})}{\mathcal{Q}(\boldsymbol{Z}_{1:L}, \tilde{\boldsymbol{Y}}_{1:R}|\boldsymbol{x})} \right]$$

$$= \mathbb{E}_{\mathcal{Q}(\boldsymbol{Z}_{1:L}, \tilde{\boldsymbol{Y}}_{1:R}|\boldsymbol{x})} \left[ \log p_{\boldsymbol{\theta}}(\boldsymbol{x}|\boldsymbol{Z}_{1:L}) - \log \frac{\mathcal{Q}(\boldsymbol{Z}_{1:L}, \tilde{\boldsymbol{Y}}_{1:R}|\boldsymbol{x})}{\mathcal{P}(\boldsymbol{Z}_{1:L}, \tilde{\boldsymbol{Y}}_{1:R})} \right]$$

$$= \mathbb{E}_{\mathcal{Q}(\boldsymbol{Z}_{1:L}, \tilde{\boldsymbol{Y}}_{1:R}|\boldsymbol{x})} \left[ \log p_{\boldsymbol{\theta}}(\boldsymbol{x}|\boldsymbol{Z}_{1:L}) - \sum_{r=1}^{R}\sum_{i=1}^{d_r} \log \frac{p_{\boldsymbol{s}_r^2}(\tilde{\boldsymbol{y}}_{r,i}|\hat{\boldsymbol{Z}}_{\ell_r})}{p_{\boldsymbol{s}_r^2}(\tilde{\boldsymbol{y}}_{r,i}|\boldsymbol{Z}_{\ell_r})} - \sum_{l=1}^{L}\sum_{i=1}^{d_l} \log \frac{\hat{P}_{s_l^2}(\boldsymbol{z}_{l,i}|\tilde{\boldsymbol{Z}}_l)}{P(\boldsymbol{z}_{l,i})} \right]$$

$$= \mathbb{E}_{\mathcal{Q}(\boldsymbol{Z}_{1:L}, \tilde{\boldsymbol{Y}}_{1:R}|\boldsymbol{x})} \left[ \log p_{\boldsymbol{\theta}}(\boldsymbol{x}|\boldsymbol{Z}_{1:L}) + \sum_{r=1}^{R}\sum_{i=1}^{d_r} \log \frac{p_{\boldsymbol{s}_r^2}(\tilde{\boldsymbol{y}}_{r,i}|\boldsymbol{Z}_{\ell_r})}{p_{\boldsymbol{s}_r^2}(\tilde{\boldsymbol{y}}_{r,i}|\hat{\boldsymbol{Z}}_{\ell_r})} + \sum_{l=1}^{L}\sum_{i=1}^{d_l} H(\hat{P}_{s_l^2}(\boldsymbol{z}_{l,i}|\tilde{\boldsymbol{Z}}_l)) - \log K_l \right],$$

$$(26)$$

where $\hat{\boldsymbol{Y}}_r = \tilde{\boldsymbol{G}}_{\boldsymbol{\phi}}^r(\boldsymbol{H}_{\boldsymbol{\phi}}^r(\boldsymbol{x}), \boldsymbol{Z}_{1:L_{r-1}})$. Since we model the dequantization process and the probabilistic decoder as Gaussians, by substituting their closed forms into the above equation, we have

$$\mathcal{J}_{\text{HQ-VAE}} = \frac{D}{2} \log \sigma^2$$

$$+ \mathbb{E}_{\mathcal{Q}(\boldsymbol{Z}_{1:L}, \tilde{\boldsymbol{Z}}_{1:L}|\boldsymbol{x})} \left[ \frac{\|\boldsymbol{x} - \boldsymbol{f}_{\boldsymbol{\theta}}(\boldsymbol{Z}_{1:L})\|_2^2}{2\sigma^2} + \sum_{r=1}^{R} \frac{\left\| \tilde{\boldsymbol{G}}_{\boldsymbol{\phi}}^r(\boldsymbol{H}_{\boldsymbol{\phi}}^r(\boldsymbol{x}), \boldsymbol{Z}_{1:L_{r-1}}) - \sum_{l\in\ell_r} \boldsymbol{Z}_l \right\|_F^2}{2\sum_{l\in\ell_r} s_l^2} - \sum_{l=1}^{L} H(\hat{P}_{s_l^2}(\boldsymbol{Z}_l|\tilde{\boldsymbol{Z}}_l)) \right],$$

$$(27)$$

where we used

$$\mathbb{E}_{\mathcal{Q}(\boldsymbol{Z}_{1:L}, \tilde{\boldsymbol{Y}}_{1:R}|\boldsymbol{x})} \left[ \frac{p_{\boldsymbol{s}_r^2}(\tilde{\boldsymbol{y}}_{r,i}|\boldsymbol{Z}_{\ell_r})}{p_{\boldsymbol{s}_r^2}(\tilde{\boldsymbol{y}}_{r,i}|\hat{\boldsymbol{Z}}_{\ell_r})} \right] = \mathbb{E}_{\mathcal{Q}(\boldsymbol{Z}_{1:L}, \tilde{\boldsymbol{Y}}_{1:R}|\boldsymbol{x})} \left[ -\frac{1}{2\sum_{l\in\ell_r} s_l^2} \|\tilde{\boldsymbol{y}}_{r,i} - \boldsymbol{y}_{r,i}\|_2^2 + \frac{1}{2\sum_{l\in\ell_r} s_l^2} \|\tilde{\boldsymbol{y}}_{r,i} - \hat{\boldsymbol{y}}_{r,i}\|_2^2 \right]$$

$$= -\mathbb{E}_{\mathcal{Q}(\boldsymbol{Z}_{1:L}, \tilde{\boldsymbol{Y}}_{1:R}|\boldsymbol{x})} \left[ \frac{1}{2\sum_{l\in\ell_r} s_l^2} \|\tilde{\boldsymbol{y}}_{r,i} - \boldsymbol{y}_{r,i}\|_2^2 \right] + \frac{d_r}{2}. \quad (28)$$

Here, we use $\tilde{\boldsymbol{Y}}_r = \hat{\boldsymbol{Y}}_r$ instead of sampling it in practical implementation.

## C  Experimental details

We explain the details of the experiments[2] in Section 5. For all the experiments except for RSQ-VAE and RQ-VAE on FFHQ and UrbanSound8K in Section 5.2, we construct architectures for both the *bottom-up* and *top-down* paths as described in Figures 1 and 8. To build these paths, we introduce two common blocks, the Resblock and Convblock by following Child (2021) as in Figure 8a, which are used in Figures 8b and 8c. Here, we denote the width and height of $\boldsymbol{H}_{\boldsymbol{\phi}}^r(\boldsymbol{x})$ as $w_r$ and $h_r$, respectively, i.e., $\boldsymbol{H}_{\boldsymbol{\phi}}^r(\boldsymbol{x}) \in \mathbb{R}^{d_b \times w_r \times h_r}$. We set $c_{\text{mid}} = 0.5$ in Figure 8. For all the experiments, we use the Adam optimizer with $\beta_1 = 0.9$ and $\beta_2 = 0.9$. Unless otherwise noted, we reduce the learning rate in half if the validation loss is not improved in the last three epochs.

In HQ-VAE, we deal with the decoder variance $\sigma^2$ using the update scheme with the maximum likelihood estimation (Takida et al., 2022a). We gradually reduce the temperature parameter of Gumbel–softmax trick with a standard scheduler $\tau = \exp(10^{-5} \cdot t)$ (Jang et al., 2017), where $t$ is the iteration step.

We set hyperparameters of VQ-VAE to standard parameter values: the balancing parameter $\beta$ in Equations (8) and (16) to 0.25, and the weight decay in EMA for the codebook update to 0.99, respectively.

---

[2] The source code is attached in the supplementary material.

Table 3: Notations of convolutional layers used in Figure 8.

| Notation | Description |
|---|---|
| $\text{Conv}_d^{(1 \times 1)}$ | 2D Convolutional layer (channel= $n$, kernel= $1 \times 1$, stride= 1, padding= 0) |
| $\text{Conv}_d^{(3 \times 3)}$ | 2D Convolutional layer (channel= $n$, kernel= $3 \times 3$, stride= 1, padding= 1) |
| $\text{Conv}_d^{(4 \times 4)}$ | 2D Convolutional layer (channel= $n$, kernel= $4 \times 4$, stride= 2, padding= 1) |
| $\text{ConvT}_d^{(3 \times 3)}$ | 2D Transpose convolutional layer (channel= $n$, kernel= $3 \times 3$, stride= 1, padding= 1) |
| $\text{ConvT}_d^{(4 \times 4)}$ | 2D Transpose convolutional layer (channel= $n$, kernel= $4 \times 4$, stride= 2, padding= 1) |

We here review the datasets used in Section 5 below.

**CIFAR10.** CIFAR10 (Krizhevsky et al., 2009) contains 10 classes of 32×32 color images, which are separated into 50,000 and 10,000 samples for train and test sets, respectively. We use the default split and further randomly select 10,000 samples from the train set to prepare the validation set.

**CelebA-HQ.** CelebA-HQ (Karras et al., 2018) contains 30,000 high-resolution face images that are selected from the CelebA dataset by following Karras et al. (2018). We use the default train/validation/test split. We preprocess the images by cropping and resizing them to the size of 256×256.

**FFHQ.** FFHQ (Karras et al., 2019) contains 70,000 high-resolution face images. In Section 5.1, we split the images into three sets: train (60,000 samples), validation (5,000 samples), and test (5,000 samples) sets. We crop and resize them to 1024×1024. In Section 5.2, we follow the same preprocessing as in Lee et al. (2022a), respectively, where it splits the images into two sets, train (60,000 samples), validation (10,000 samples) sets and crop and resize them to 256×256.

**ImageNet.** ImageNet (Deng et al., 2009) contains 1000 classes of natural images in RGB scales. We use the default train/val/test split. We crop and resize the images to 256×256.

**UrbanSound8K.** UrbanSound8K (Salamon et al., 2014) contains 8,732 labeled audio clips of urban sound from 10 classes. UrbanSound8K has a wide range of sound classes, such as dog barking and drilling. UrbanSound8K is divided into 10 folds, and we use the fold 1-8/9/10 as the train/validation/test split. The duration of each audio clip is less than 4 seconds. In our experiments, to align the length of input audio, we pad the all audio clips to 4 seconds. We also convert the all audio clips to 16 bit and down-sampled them to 22,050 kHz. A 4-second waveform audio clip is converted to a Mel spectrogram with shape $80 \times 344$. We preprocess an audio clip following the paper (Liu et al., 2021):

1. We extract an 80-dimensional Mel spectrogram using the short-time Fourier transform (STFT) with a frame size of 1024, a hop size of 256, and a Hann window.
2. We apply dynamic range compression to the Mel spectrogram by first clipping it to a minimum value of $1 \times 10^{-5}$ and then applying a logarithmic transformation.

### C.1 SQ-VAE-2 vs VQ-VAE-2

### C.1.1 Comparison on CIFAR10 and CelebA-HQ

We construct the architecture as depicted in Figures 1 and 8. To build the *top-down* paths, we use two *injected top-down* layers (i.e., $R = 2$) with $w_1 = h_1 = 8$ and $w_2 = h_2 = 16$ for CIFAR10, and three layers (i.e., $R = 3$) with $w_1 = h_1 = 8$, $w_2 = h_2 = 16$ and $w_3 = h_3 = 32$ for CelebA-HQ, respectively. For the *bottom-up* paths, we repeatedly stack two Resblocks and an average pooling layer once and four times, respectively, for CIFAR10 and CelebA-HQ. We set the learning rate to 0.001 and train all the models for a maximum of 100 epochs with a mini-batch size of 32.

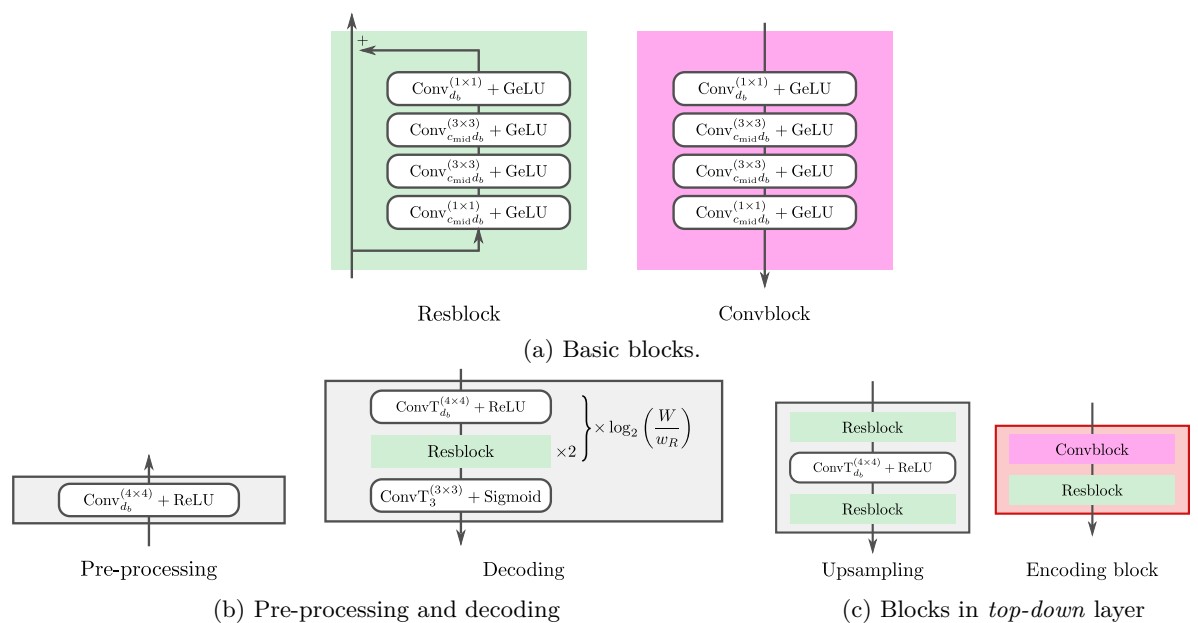

Figure 8: Architecture details in Figure 1. Notations of convolutional layers, $\text{Conv}_d^{(k \times k)}$ and $\text{ConvT}_d^{(k \times k)}$, are summarized in Table 3.

#### C.1.2 Comparison on large-scale datasets

We construct the architecture as depicted in Figures 1 and 8. To build the *top-down* paths, we use two *injected top-down* layers (i.e., $R = 2$), with $w_1 = h_1 = 32$ and $w_2 = h_2 = 64$ for ImageNet, and three layers (i.e., $R = 3$) with $w_1 = h_1 = 32$, $w_2 = h_2 = 64$ and $w_3 = h_3 = 128$ for FFHQ, respectively. For the *bottom-up* paths, we repeatedly stack two Resblocks and an average pooling layer three times and five times respectively for ImageNet and FFHQ. We set the learning rate to 0.0005. We train ImageNet and FFHQ for a maximum of 50 and 200 epochs with a mini-batch size of 512 and 128, respectively. Figure 9 and Figure 10 show reconstructed samples of SQ-VAE-2 on ImageNet and FFHQ, respectively.

### C.2 RSQ-VAE vs RQ-VAE

#### C.2.1 Comparison on CIFAR10 and CelebA-HQ

We construct the architecture as depicted in Figures 1 and 8 without *injected top-down* layers, i.e., $R = 1$. We set the resolution of $\boldsymbol{H}_\phi(\boldsymbol{x})$ to $w = h = 8$. For the *bottom-up* paths, we repeatedly stack two Resblocks and an average pooling layer once and four times, respectively, for CIFAR10 and CelebA-HQ. We set the learning rate to 0.001 and train all the models for a maximum of 100 epochs with a mini-batch size of 32.

#### C.2.2 Improvement in perceptual quality

In this experiment, we use the same network architecture as that used in Lee et al. (2022a). We set the learning rate to 0.001 and train an RSQ-VAE model for a maximum of 300 epochs with a mini-batch size of 128 (4 GPUs, 32 samples for each GPU) on FFHQ. We use our modified LPIPS loss (see Appendix C.4) in training. For evaluation, we compute rFID scores with the code provided in their repository[3] on the validation set (10,000 samples). And, we use the pre-trained RQ-VAE model offered in the same repository for evaluating RQ-VAE.

We show examples of reconstructed images in Appendix C.4 after we explain our modified LPIPS loss.

---

[3]https://github.com/kakaobrain/rq-vae-transformer

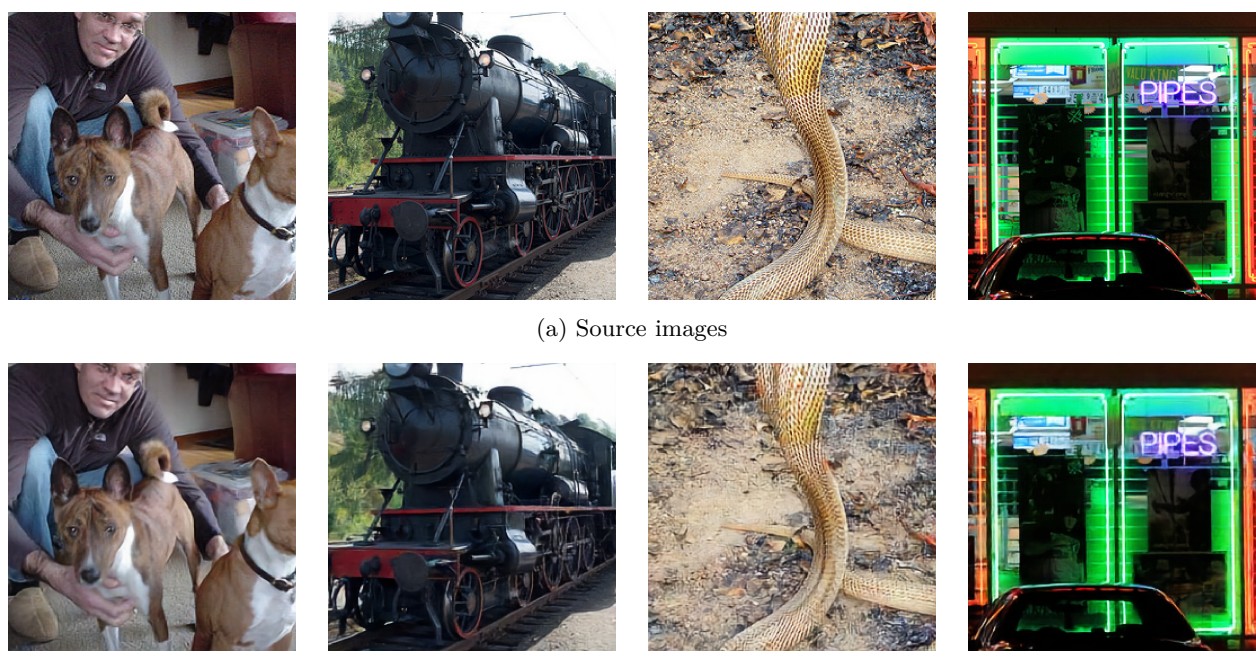

(a) Source images

(b) Reconstructed images

Figure 9: Reconstructed samples of SQ-VAE-2 trained on ImageNet

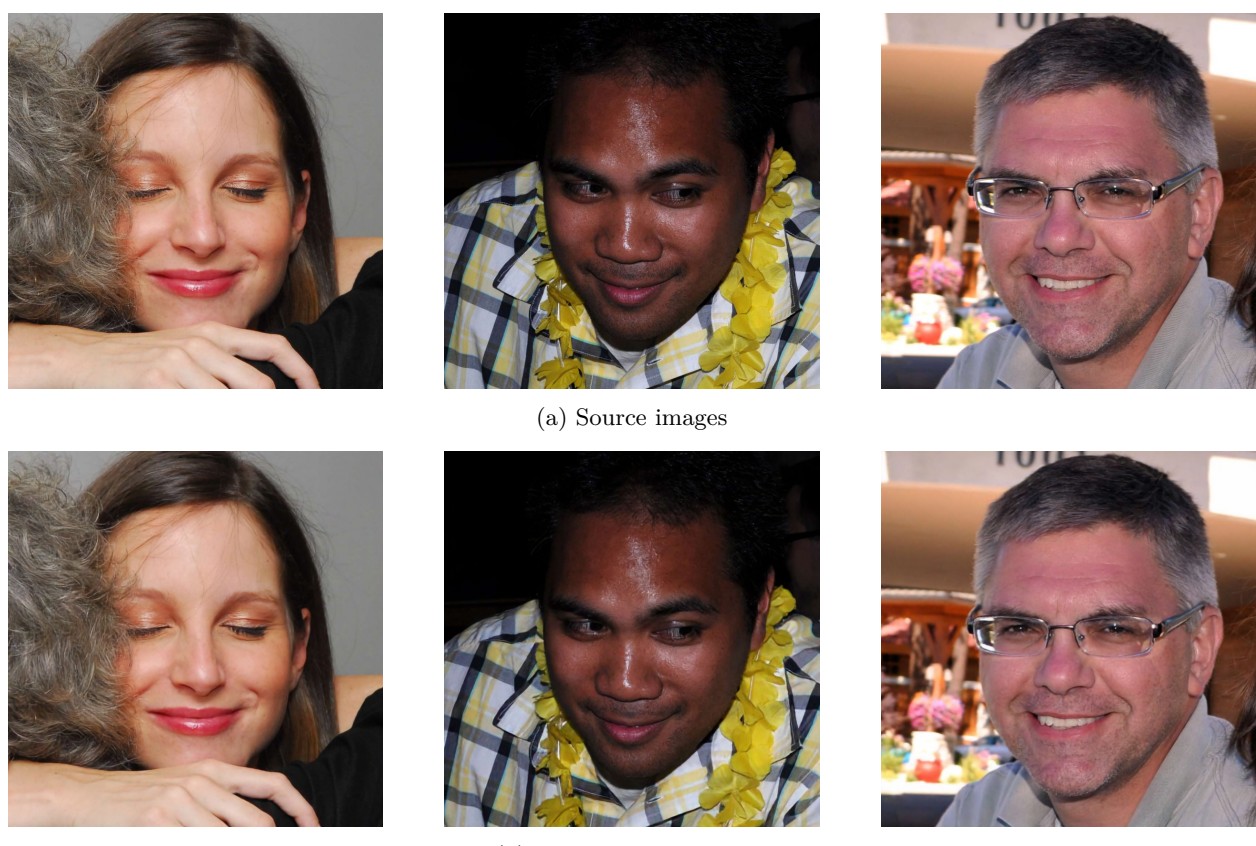

(a) Source images

(b) Reconstructed images

Figure 10: Reconstructed samples of SQ-VAE-2 trained on FFHQ

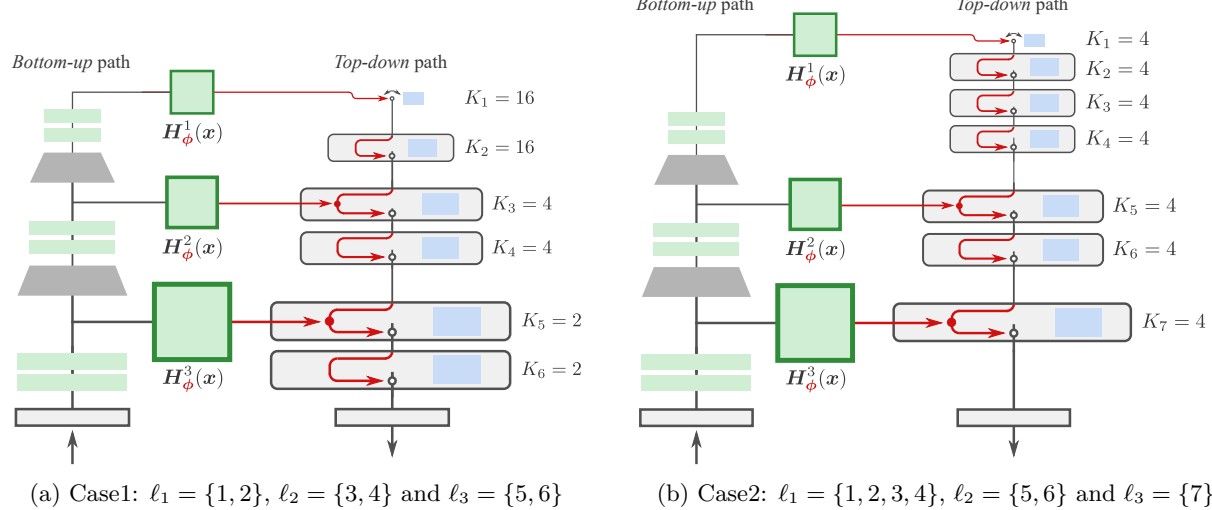

(a) Case1: $\ell_1 = \{1, 2\}$, $\ell_2 = \{3, 4\}$ and $\ell_3 = \{5, 6\}$  (b) Case2: $\ell_1 = \{1, 2, 3, 4\}$, $\ell_2 = \{5, 6\}$ and $\ell_3 = \{7\}$

Figure 11: Architecture of the hybrid model in Appendix C.3.

### C.2.3 Validation on an audio dataset

We construct the architecture by following the previous audio generation work (Liu et al., 2021). For the *top-down* paths, the architecture consists of several strided convolutional layers in parallel (Xian et al., 2021). We use four strided convolutional layers consisting of two sub-layers with stride 2, followed by two ResBlocks with ReLU activations. The kernel sizes of these four strided convolutional layers are $2 \times 2$, $4 \times 4$, $6 \times 6$ and $8 \times 8$ respectively. We add the outputs of the four strided convolutional layers, and pass it to a convolutional layer with kernel size $3 \times 3$. Then we get the resolution of $\boldsymbol{H_\phi(x)}$ to $w = 20, h = 86$. For the *bottom-up* paths, we stack a convolutional layer with kernel size $3 \times 3$, two Resblocks with ReLU activations, and two transposed convolutional layers with stride 2 and kernel size $4 \times 4$. We set the learning rate to 0.001 and train all the models for a maximum of 100 epochs with a mini-batch size of 32. To convert the normalized log-Mel spectrogram to the waveform, we use the same HiFi-GAN vocoder (Kong et al., 2020) as used in the audio generation work (Liu et al., 2021). The HiFi-GAN vocoder is trained on the train set of UrbanSound8K from scratch (Liu et al., 2021). The post-processing of assessors follows the paper (Series, 2014): assessors are excluded from the aggregated scores if they rate the hidden reference for more than 15% of the test signals lower than a score of 90.

As an example of demonstration, we randomly select audio clips from our test split of UrbanSound8K and show their reconstructed Mel spectrogram samples from RQ-VAE and RSQ-VAE in Figure 12. While the samples from RQ-VAE have difficulty to reconstruct the sources with shared codebooks, the samples from RSQ-VAE reconstruct detailed features of the sources.

### C.3 Empirical study of top-down layers

For a demonstration, we build two HQ-VAEs by combinatorially using both the *injected top-down* and the *residual top-down* layers with three resolutions, $w_1 = h_1 = 16$, $w_2 = h_2 = 32$ and $w_3 = h_3 = 64$. We construct the architectures as described in Figure 11 and train them on CelebA-HQ. Figure 14 shows the progressively reconstructed images for each case. We can observe the same tendencies as in Figures 5a and 5c.

### C.4 Perceptual loss for images

We found that LPIPS loss (Zhang et al., 2018), which is a perceptual loss for images (Johnson et al., 2016), works well with our HQ-VAE. However, we also noticed that just replacing $\|\boldsymbol{x} - \boldsymbol{f_\theta}(\boldsymbol{Z}_{1:L})\|_2^2$ in the objective function of HQ-VAE (Equations (7) and (15)) with an LPIPS loss $\mathcal{L}_{\text{LPIPS}}(\boldsymbol{x}, \boldsymbol{f_\theta}(\boldsymbol{Z}_{1:L}))$ leads to artifacts in

generated images. We hypothesize that those artifacts are caused by the max-pooling layers in VGGNet used in LPIPS. Signals from VGGNet might not reach all pixels in backpropagation due to the max-pooling layers. To mitigate this issue, we applied a padding-and-trimming operation to both a generated image $f_{\theta}(Z_{1:L})$ and the corresponding reference image $x$ before the LPIPS loss function. That is $\mathcal{L}_{\text{LPIPS}}(\text{pt}\,[x]\,,\text{pt}\,[f_{\theta}(Z_{1:L})])$, where pt [ ] denotes our padding-and-trimming operator. The PyTorch implementation of such an operation is described below.

```python
import random
import torch
import torch.nn.functional as F

def padding_and_trimming(
    x_rec, # decoder output
    x  # reference image
):
    _, _, H, W = x.size()

    x_rec = F.pad(x_rec, (15, 15, 15, 15), mode='replicate')
    x = F.pad(x, (15, 15, 15, 15), mode='replicate')

    _, _, H_pad, W_pad = x.size()
    top = random.randrange(0, 16)
    bottom = H_pad - random.randrange(0, 16)
    left = random.randrange(0, 16)
    right = W_pad - random.randrange(0, 16)

    x_rec = F.interpolate(x_rec[:, :, top:bottom, left:right],
                          size=(H, W), mode='bicubic', align_corners=False)
    x = F.interpolate(x[:, :, top:bottom, left:right],
                      size=(H, W), mode='bicubic', align_corners=False)

    return x_rec, x
```

Note that our padding-and-trimming operation includes downsampling with a random ratio. We assume that this random downsampling provides a generative model with diversified signals in backpropagation across training iterations, which makes the model more generalizable.

Figure 13 shows images reconstructed by an RSQ-VAE model trained with a normal LPIPS loss, $\mathcal{L}_{\text{LPIPS}}(x, f_{\theta}(Z_{1:L}))$, and ones reconstructed by an RSQ-VAE model trained with our modified LPIPS loss, $\mathcal{L}_{\text{LPIPS}}(\text{pt}\,[x]\,,\text{pt}\,[f_{\theta}(Z_{1:L})])$. As shown, our padding-and-trimming technique alleviates the artifacts issue. For example, vertical line noise can be seen in hairs in the images generated by the former model, but those lines are removed or softened in the images generated by the latter model. Indeed, our technique improves rFID from 10.07 to 8.47.

## C.5 Progressive coding

For demonstration purpose of Figure 5a, we incorporate the concept of *progressive coding* (Ho et al., 2020; Shu & Ermon, 2022) to our framework, which helps hierarchical models to be more sophisticated in progressive lossy compressing and may generate high-fidelity samples. One can train SQ-VAE-2 to achieve progressive lossy compression (as in Figure 5a) by introducing additional generative processes $\tilde{x}_l \sim \mathcal{N}(\tilde{x}_l; f_{\theta}(Z_{1:l}), \sigma_l^2 I)$ for $l \in [L]$. We here derive the corresponding ELBO objective with this concept. Its benefit is to produce more reasonable reconstructed images only with higher layers (i.e., using only low-resolution information $H_{\phi}^r(x)$).

First, we consider corrupted data $\tilde{x}_l$ for $l \in [L]$, which is obtained by adding noises, for example, i.e., $\tilde{x}_l = x + \epsilon_l$. We here adopt the Gaussian distribution $\epsilon_{l.d} \sim \mathcal{N}(0, v_l)$ for the noises. Note that $\{\sigma_l^2\}_{l=1}^L$ is set to be a non-increasing sequence. We model the generative process using only the top $l$ groups as

Table 4: FID scores on FFHQ (256×256). The values with † represents the scores reported in Lee et al. (2022a)

| Model | FID ↓ |
|---|---|
| Very deep VAE (Child, 2021) | $28.5^\dagger$ |
| VQ-GAN + Transformer (Esser et al., 2021b) | $11.4^\dagger$ |
| RQ-VAE + RQ-Transformer Lee et al. (2022a) | $10.38^\dagger$ |
| RSQ-VAE + RQ-Transformer | 9.74 |
| RSQ-VAE + contextual RQ-Transformer | 8.46 |

$p_{\boldsymbol{\theta}}^l(\tilde{\boldsymbol{x}}_l) = \mathcal{N}(\tilde{\boldsymbol{x}}_l; f_{\boldsymbol{\theta}}(\boldsymbol{Z}_{1:l}), \sigma_l^2 \boldsymbol{I})$. Now the ELBO is obtained as

$$\mathcal{J}_{\text{SQ-VAE-2}}^{\text{prog}} = \sum_{l=1}^L \frac{D}{2} \log \sigma_l^2 + \mathbb{E}_{\mathcal{Q}(\boldsymbol{Z}_{1:L}, \tilde{\boldsymbol{z}}_{1:L}|\boldsymbol{x})} \left[ \frac{\|\boldsymbol{x} - f_{\boldsymbol{\theta}}(\boldsymbol{Z}_{1:l}) + Dv_l\|_2^2}{2\sigma_l^2} + \frac{\|\tilde{\boldsymbol{Z}}_l - \boldsymbol{Z}_l\|_F^2}{2s_l^2} - H(\hat{P}_{s_l^2}(\boldsymbol{Z}_l|\tilde{\boldsymbol{Z}}_l)) \right] \quad (29)$$

In Section 5.3, we simply set $v_l = 0$ in the above objective when this technique is activated.

This concept can be also applied to the hybrid model derived in Appendix B by considering additional generative processes $p_{\boldsymbol{\theta}}^r(\tilde{\boldsymbol{x}}_r) = \mathcal{N}(\tilde{\boldsymbol{x}}_r; f_{\boldsymbol{\theta}}(\boldsymbol{Z}_{1:L_r}), \sigma_r^2 \boldsymbol{I})$. The ELBO objective is as follows:

$$\mathcal{J}_{\text{HQ-VAE}}^{\text{prog}} = \sum_{r=1}^R \frac{D}{2} \log \sigma_r^2$$

$$+ \mathbb{E}_{\mathcal{Q}(\boldsymbol{Z}_{1:L}, \tilde{\boldsymbol{Y}}_{1:R}|\boldsymbol{x})} \left[ \sum_{r=1}^R \frac{\|\boldsymbol{x} - \boldsymbol{f}_{\boldsymbol{\theta}}(\boldsymbol{Z}_{1:L_r}) + Dv_r\|_2^2}{2\sigma_r^2} + \sum_{r=1}^R \frac{\left\|\hat{\boldsymbol{Y}}_r - \sum_{l \in \ell_r} \boldsymbol{Z}_l\right\|_F^2}{2\sum_{l \in \ell_r} s_l^2} - \sum_{l=1}^L H(\hat{P}_{s_l^2}(\boldsymbol{Z}_l|\tilde{\boldsymbol{Z}}_l)) \right]. \quad (30)$$

In Section C.3, we simply set $v_l = 0$ in the above objective when this technique is activated.

# D   Application of HQ-VAE to image generation

To demonstrate the applicalability of HQ-VAE to generation tasks, we train two prior models, an RQ-Transformer (Lee et al., 2022a) and a contextual RQ-Transformer (Lee et al., 2022b), on the FFHQ latent features extracted by RSQ-VAE. We numerically compare our RSQ-VAE-based generative models with the current VAE models, very deep VAE (Child, 2021), VQ-GAN (Esser et al., 2021b) and RQ-VAE (Lee et al., 2022a). We note that VQ-GAN and RQ-VAE are based on the same architecture and their training schemes use adversarial training with a discriminator. We calculate FID score for our models to evaluate the generation performance. As shown in Table 4, RSQ-VAE with RQ-Transformer achieves better generation performance than VQ-GAN with Transformer, and RQ-VAE with RQ-Transformer despite that RSQ-VAE is trained without adversarial training. Figure 15 shows the generated samples from RSQ-VAE with the contextual Transformer. According to the results, the latent representations learned by HQ-VAEs are shown to be tractable for prior models.

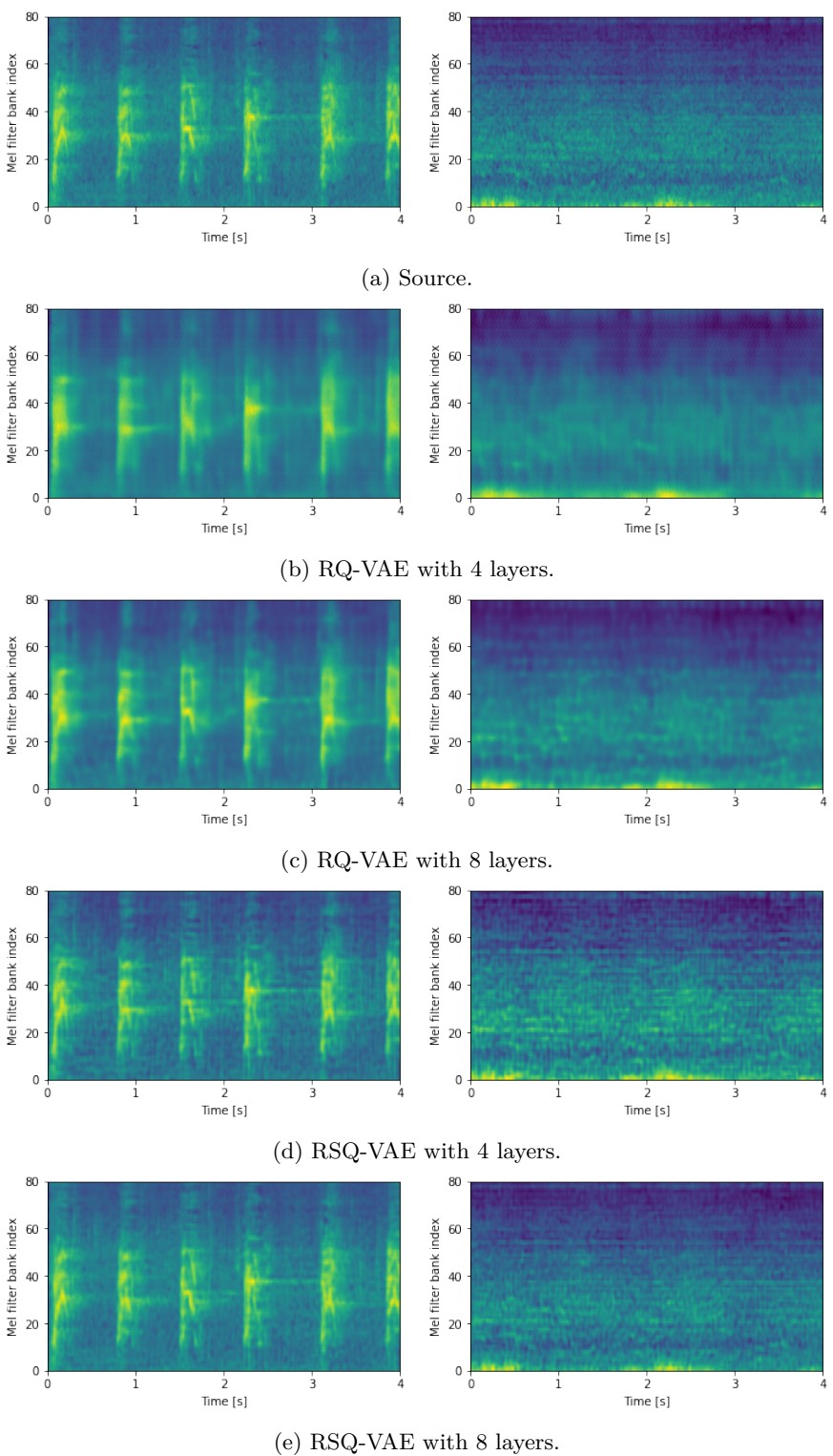

(a) Source.

(b) RQ-VAE with 4 layers.

(c) RQ-VAE with 8 layers.

(d) RSQ-VAE with 4 layers.

(e) RSQ-VAE with 8 layers.

Figure 12: Mel spectrogram of (a) sources and (b)-(e) reconstructed samples of UrbanSound8K dataset. The left panel and the right panel are audio clips of dog barking and drilling, respectively. We observe that RQ-VAEs struggle to reconstruct the sources with shared codebooks. In contrast, the reconstruction of RSQ-VAE can reflect the details of the source samples.

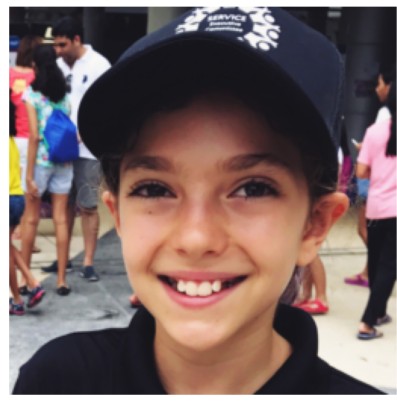 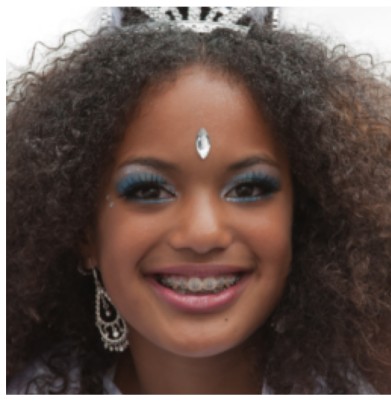 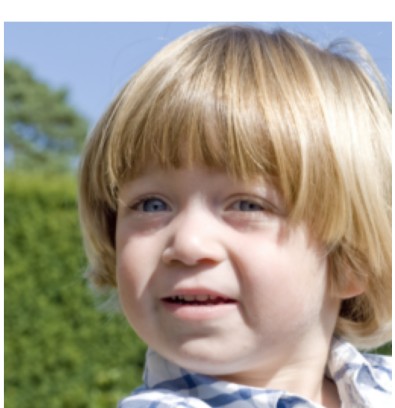

(a) Source

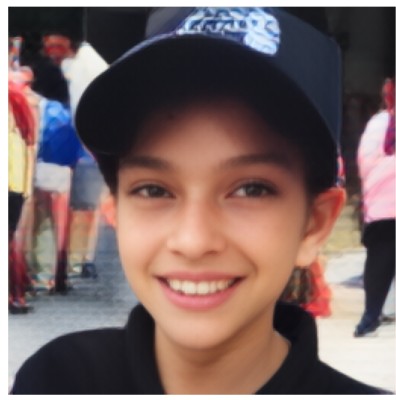 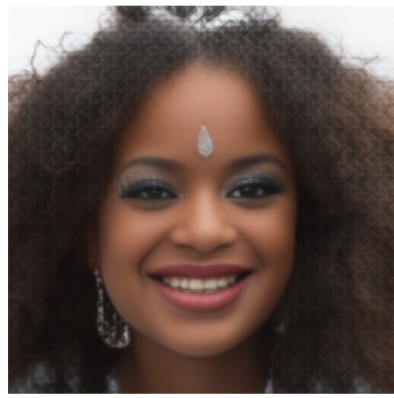 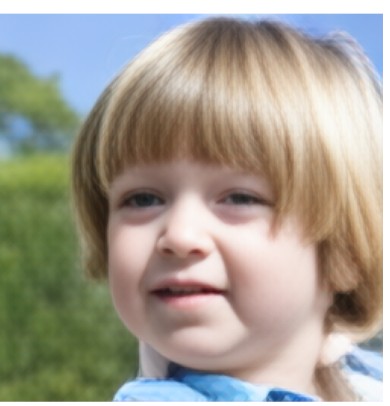

(b) RSQ-VAE trained with a normal LPIPS loss (rFID= 10.07)

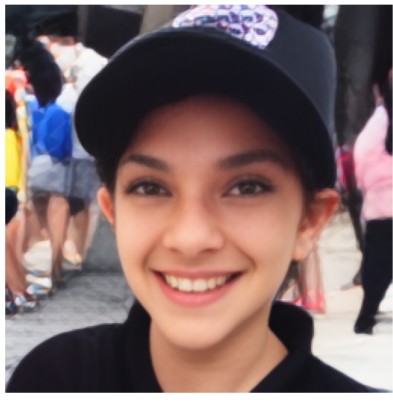 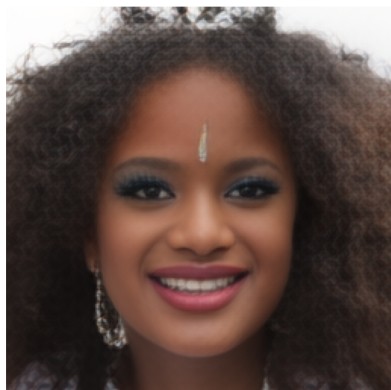 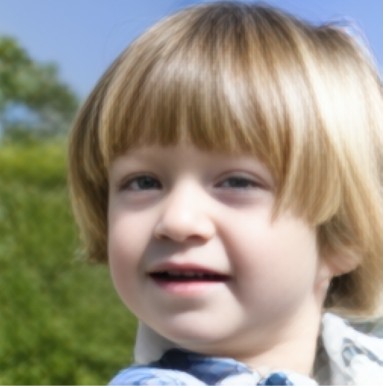

(c) RSQ-VAE trained with our improved LPIPS loss (rFID= 8.47)

Figure 13: Reconstructed samples of FFHQ.

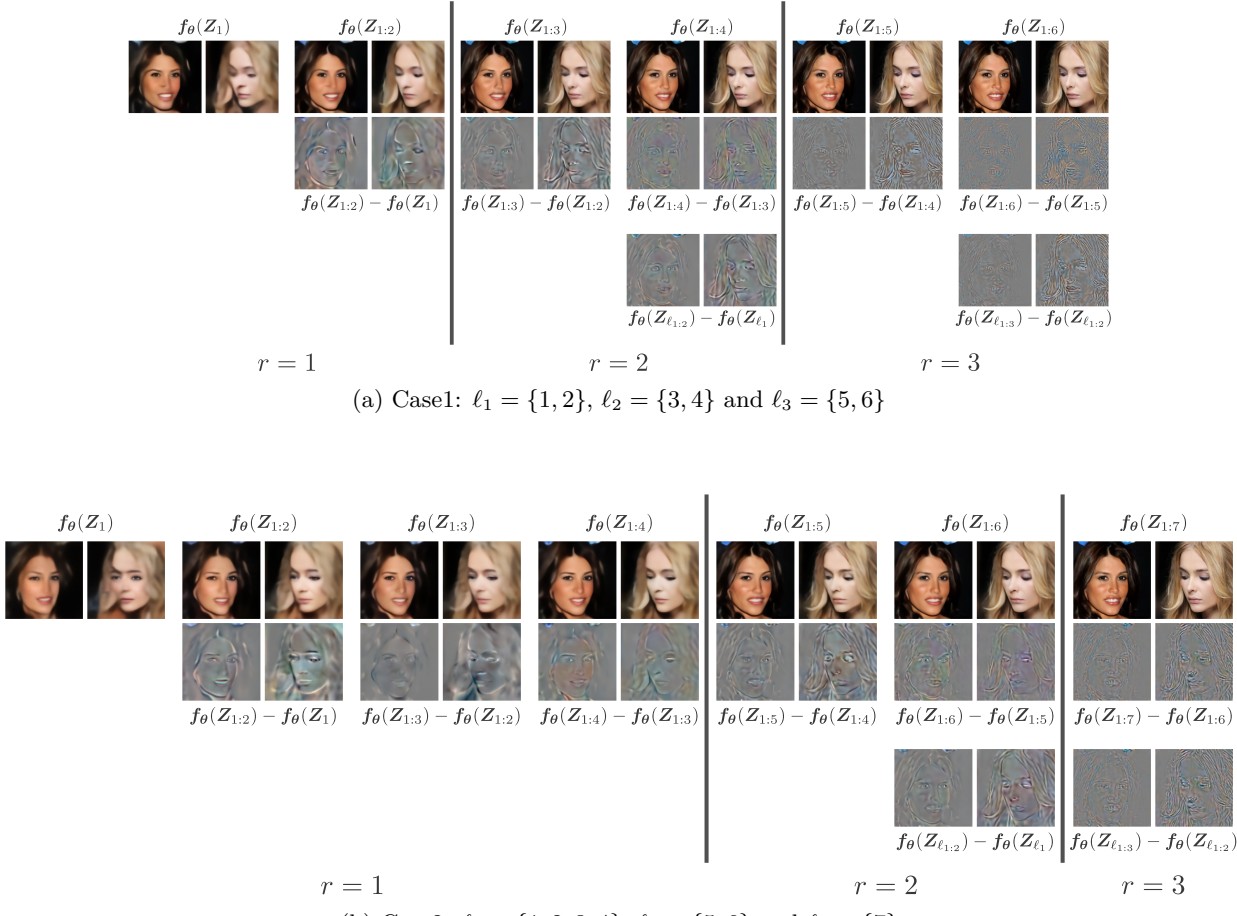

(a) Case1: $\ell_1 = \{1, 2\}$, $\ell_2 = \{3, 4\}$ and $\ell_3 = \{5, 6\}$

(b) Case2: $\ell_1 = \{1, 2, 3, 4\}$, $\ell_2 = \{5, 6\}$ and $\ell_3 = \{7\}$

Figure 14: Reconstructed images and magnified differences of HQ-VAE on CelebA-HQ

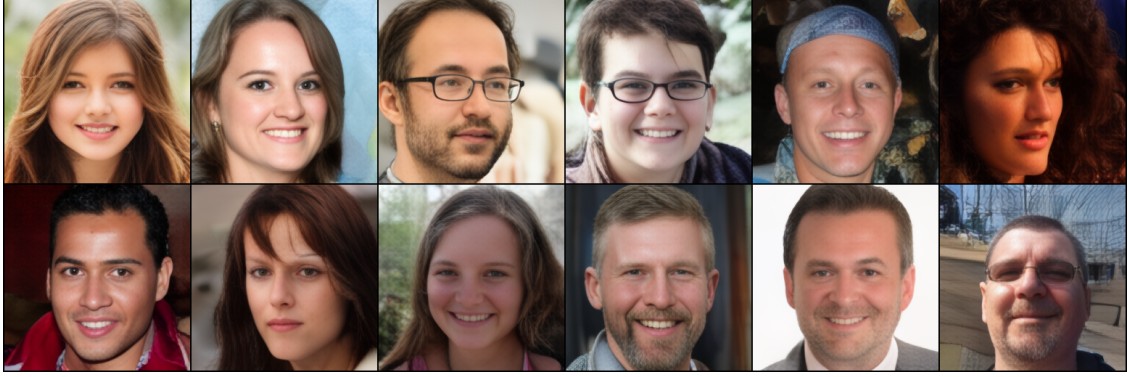

Figure 15: Samples of FFHQ from RSQ-VAE with contextual RQ-Transformer (Lee et al., 2022b).

