# OpenReview forum: "HQ-VAE: Hierarchical Discrete Representation Learning with Variational Bayes"
_TMLR — Rejected by TMLR_

### Review · Reviewer_nVD1 · 2023-04-16

**Summary Of Contributions:**

This paper proposes a new model for hierarchical extensions of VQ-VAEs, which they claim to alleviate some of the instable training and poor reconstruction accuracy issues that previously existed. Their proposed approach stochastically learns hierarchical discrete representations which results in a natural hierarchical model with improved performance and more stable training. The authors experiment with their model for image and audio generation and show some promising empirical results.

**Audience:**

Yes

**Claims And Evidence:**

No

**Requested Changes:**

1. Clarify the claims around 'stable training' and 'improved reconstruction', perhaps by adding more experiments.
2. The changes requested to Table 1, Figure 2, Figure 3, etc.
3. Better summary of takeaway messages.

**Strengths And Weaknesses:**

Strengths:
1. The paper is well motivated.
2. The method seems to be well formulated via extensions of the variational bayes framework for hierarchical latent variables, and the derivations of objective functions seem to be right.
3. Paper is generally well written

Weaknesses:
1. The biggest weakness is that there is no solid justification of the claimed advantages of the method. The authors claim 'more stable training' in the abstract, but there are no comments on how you would quantitatively evaluate this property and that the proposed method indeed does better according to this metric.
2. Also the authors claim improved reconstruction performance - this is better evaluated but it is unclear if measuring reconstruction via MSE-type metrics is most appropriate - there should be at least human evaluation to check the actual quality of generated content.
3. In some cases it is also unclear how much the MSE improvement is between proposed method and previous baselines, take Figure 2 for example. Error bars for Figure 2 would also help.
4. In Table 1 the columns under 'Codebook perplexity' and Z1, Z2, Z3 are not clear what they mean. There's also a missing entry for Imagenet.
5. Figure 3 comparisons should include at least some existing methods - as far as I can tell they are all ablations and variations of their proposed method.
6. The final takeaway messages can be sharpened. The authors propose quite a few variations of their method but a final guide for which method to use in which situation can help, based on their experiments.

---

> ### Author Response · Authors · 2023-05-23
>
> Thank you for your valuable comments.
>
> > (W1) The biggest weakness is that there is no solid justification of the claimed advantages of the method. The authors claim 'more stable training' in the abstract, but there are no comments on how you would quantitatively evaluate this property and that the proposed method indeed does better according to this metric.
>
> In the abstract, we used the term “stabilize” to specify mitigating codebook/layer collapse issue to enhance the reconstruction performance. We will modify the abstract to clarify the meaning of “stabilize”. We used codebook perplexity to check if the codebook collapse occurred in the experiments (please see Table 1 and Figure 3(c)).
>
>
> ---
>
> > (W2) Also the authors claim improved reconstruction performance - this is better evaluated but it is unclear if measuring reconstruction via MSE-type metrics is most appropriate - there should be at least human evaluation to check the actual quality of generated content.
>
> We agree that the use of MSE-type metrics (based on Euclidean distance between the original and reconstructed samples) solely cannot assess the actual reconstruction performance well.
>
> In response to this comment, we conducted human evaluation to check the perceptual quality of reconstructed samples in audio dataset by following MUSHRA tests (ITU-R BS2001, Schoeffler2018) because there was no perceptual metric for the experiment on audio dataset. For this experiment, we confirmed that RSQ-VAE outperforms RQ-VAE even in terms of perceptual reconstruction. We will add the results to Section 5.2.
>
> In contrast, for the experiment on vision dataset, we used LPIPS, SSIM and rFID besides RMSE to evaluate the reconstruction performance in the original manuscript. LPIPS and SSIM are commonly used to approximately evaluate the perceptual difference between original and reconstructed images (Wang+2004, Zhang+2018). The metric rFID measures distributional difference between the original and reconstructed images. Furthermore, we can see that our models outperformed the baselines even in the sense of these perceptual metrics. We will add the explanation on these metrics to the manuscript.
>
> [ITU-R BS2001] “Method for the subjective assessment of intermediate sound quality (MUSHRA)”
>
> [Wang+2004] “The unreasonable effectiveness of deep features as a perceptual metric”
>
> [Schoeffler2018] “webMUSHRA — A comprehensive framework for web-based listening tests”
>
> [Zhang+2018] “Image quality assessment: from error visibility to structural similarity”
>
>
> ---
>
> > (W3) In some cases it is also unclear how much the MSE improvement is between proposed method and previous baselines, take Figure 2 for example. Error bars for Figure 2 would also help.
>
> Thank you for the suggestion. We will add the error bars in Figures 2 and 3. We can see the performance improvement in the plot of the metrics, LPIPS and SSIM, especially in the lower codebook size case.
>
>
> ---
>
> > (W4) In Table 1 the columns under 'Codebook perplexity' and Z1, Z2, Z3 are not clear what they mean. There's also a missing entry for Imagenet.
>
> Codebook perplexity in the context of VQ-based compression is defined as the exponential of the entropy of the aggregated categorical posterior $Q(Z_l)$, which indicates how the codebook elements are evenly used for the discrete representation (Roy+2018, Takida+2022, ). As one of the extreme cases, the perplexity value becomes 1 if the quantizer always select a certain code vector. As the other extreme case, the perplexity becomes the number of codes, which is the maximum value, if $Q(Z_l)$ is a uniform categorical distribution.
>
> We calculated codebook perplexity for each layer to check if layer collapse occurred per layer. In section 5.1, we introduced two *injected top-down* layers by following the experimental setting in [Razavi+2019] for ImageNet (and three *injected top-down* layers for FFHQ). That is why Table 1 does not contain the codebook perplexity for $Z_3$ in the ImageNet case. We can see that the VQ-VAE-2 models rely heavily on the bottom layers in Table 1.
>
> We will add the definition of codebook perplexity and the above explanation to Section 5.1.
>
> [Roy+2018] “Theory and experiments on vector quantized autoencoders”
>
> [Takida+2022] “SQ-VAE: Variational bayes on discrete representation with self-annealed stochastic quantization”

---

> > ### Author Response · Authors · 2023-05-23
> >
> > > (W5) Figure 3 comparisons should include at least some existing methods - as far as I can tell they are all ablations and variations of their proposed method.
> >
> > In this figure, we compared our proposed model RSQ-VAE with the baseline model RQ-VAE, which is the only existing discrete representation learning method based on residual vector quantization to our best knowledge. RQ-VAE is deterministic model similarly to VQ-VAE, and may suffer from codebook collapse. So, RQ-VAE has been often used with some useful techniques to mitigate the issue, and we incorporated them into our empirical comparison. [Lee+2022] and [Défossez+2022] adopted code reset, which is also known as random restart, to mitigate the codebook collapse issue. So, we added RQ-VAE with codebook reset to the baseline. Furthermore, [Lee+2022] proposed to share a codebook for all the layers to enhance the utility of its codes. Since this technique is applicable to RSQ-VAE without destroying the variational Bayes framework, we compare RSQ-VAE and RQ-VAE even in the case where codebook is shared for all the layers. We will revise the corresponding part in Section 5.2 to clarify which models are our proposal.
> >
> >
> > ---
> >
> > > (W6) The final takeaway messages can be sharpened. The authors propose quite a few variations of their method but a final guide for which method to use in which situation can help, based on their experiments.
> >
> > Thank you for the suggestion. Putting the final takeaway messages would make the applications of our models clearer. We summarized our takeaway messages while conducting an additional experiment. Please refer to our reply posted in the thread “To all reviewers”.
> >
> > We will add an additional experiment to Sections 5.3 and discuss the takeaway messages by creating a new section (Section 6.2) in the manuscript.

---

> > > ### Author Response · Authors · 2023-05-26
> > > **Manuscript revision**
> > >
> > > We thank you again for your constructive feedback. We have revised the manuscript accordingly. We highlighted the revised parts in red.
> > >
> > > > (RC1) Clarify the claims around 'stable training' and 'improved reconstruction', perhaps by adding more experiments.
> > >
> > > We have modified the abstract and Sections 5.1, and added the MUSHURA experiment to Section 5.2 according to our comments on Weaknesses 1 and 2.
> > >
> > >
> > > ---
> > >
> > > > (RC2) The changes requested to Table 1, Figure 2, Figure 3, etc.
> > >
> > > We have reflected the suggestion on Table 1, Figures 2, 3, and Sections 5.1, 5.2 according to our comments on Weaknesses 3, 4 and 5. We note that some error bars in Figures 2 and 3 are imperceptible due to small deviations (we can see similar tendencies, i.e., small deviations, in Tables 1 and 2).
> > >
> > >
> > > ---
> > >
> > > > (RC3) Better summary of takeaway messages.
> > >
> > > We have created Section 6.2 and put the takeaway messages discussed in Weakness 6 in the new section. We have also added an empirical comparison of SQ-VAE-2 and RSQ-VAE to Section 5.3.

---

### Review · Reviewer_P5Kq · 2023-04-17

**Summary Of Contributions:**

The paper proposes an extension of the the recently proposed SQ-VAE (stochastically-quantized VAE) (Takida 2022). This results in HQ-VAE, a hierarchical VAE using bidrectional inference similar to ladder VAE (Sonderby 2016) and ResNet-VAE (Kingma 2016). The performance of HQ-VAE is empirically validated and shown to achieve lower reconstruction error and have more entropic latents than hierarchical VQ-VAEs.


-----
Yuhta Takida, Takashi Shibuya, WeiHsiang Liao, Chieh-Hsin Lai, Junki Ohmura, Toshimitsu Uesaka, Naoki
Murata, Takahashi Shusuke, Toshiyuki Kumakura, and Yuki Mitsufuji. SQ-VAE: Variational bayes on
discrete representation with self-annealed stochastic quantization. In Proc. International Conference on
Machine Learning (ICML), 2022.

Casper Kaae Sønderby, Tapani Raiko, Lars Maaløe, Søren Kaae Sønderby, and Ole Winther. Ladder variational autoencoders. In Proc. Advances in Neural Information Processing Systems (NeurIPS), pp. 3738–
3746, 2016.

Kingma, Durk P., et al. "Improved variational inference with inverse autoregressive flow." Advances in neural information processing systems 29 (2016).

**Audience:**

Yes

**Claims And Evidence:**

No

**Requested Changes:**

The paper needs to be more clear about its scope of contribution, given it's a straightforward extension of SQ-VAE and does not make significant methodological contribution.
The claim to novelty -- "we propose a novel framework to stochastically learn hierarchical discrete representation on the basis of the variational Bayes framework" -- is thus problematic, as everything except the "hierarchical" part is existing work (SQ-VAE), and even the "hierarchical" part is fairly standard from literature.

Additionally, a more careful comparison of SQ-VAE with the existing probabilistic formulations of  Sonderby 2017 and Singh 2022 would be insightful and add to the contributions, given the original SQ-VAE paper did not address these two overlapping prior works.

The empirical comparison should also be based on R-D curves (training models with different weightings between rate and distortion losses) to be correct.

Also see my comments about weaknesses above.

----
Sønderby, Casper Kaae, Ben Poole, and Andriy Mnih. "Continuous relaxation training of discrete latent variable image models." Beysian DeepLearning workshop, NIPS. Vol. 201. 2017.

Singh, Gautam, Fei Deng, and Sungjin Ahn. "Illiterate dall-e learns to compose." arXiv preprint arXiv:2110.11405 (2021).



**Strengths And Weaknesses:**

Strength:

- The exposition is mostly clear and straightforward to follow.

Weakness:

- The current work is mostly about an extended architecture of previous work (SQ-VAE), and the hierarchical architecture itself is directly adapted from literature (ResNet-VAE, VQ-VAE2), thus there is very little methodological contribution from a machine learning perspective.

- The proposed advantages of HQ-VAE are mostly a rehash of those of SQ-VAE: ("(1) greatly
reduces the number of hyperparameters to be tuned (only the one from the Gumbel-softmax trick), and (2) enhances codebook usage without any heuristics thanks to the self-annealing effect"). Given that the present work builds on SQ-VAE, it is insufficient to just demonstrate the benefit of SQ-VAE (and compare with the deterministically quantized VQ-VAE and variants), but it should rather focus on what advantages a hierarchical extension brings in addition to SQ-VAE.

- If the focus was rather on the benefit of the probabilistic quantization aspect of HQ-VAE, then there is insufficient justification given there are existing probabilistic VQ-VAE formulations which are closely related to (or even subsume) HQ-VAE/SQ-VAE, such as Sonderby 2017 and discrete VAE (dVAE) (Singh 2022). A more detailed discussion and comparison would be needed.

- The methodology for comparing the models seems problematic. The comparison based on reconstruction error / latent perplexity can be misleading and inclusive; it's more principled to compare rate-distortion curves like in Williams 2020. Having lower reconstruction error AND higher latent perplexity does not necessarily mean a better model; a VAE can in theory be tuned to operate anywhere on the R-D curve such that it operates at lower distortion and higher rate (Alemi 2018). Thus the conclusion that the HQ-VAE attains lower distortion / higher latent perplexity does not necessarily mean it's "better" than its deterministically quantized counterpart.


----
Sønderby, Casper Kaae, Ben Poole, and Andriy Mnih. "Continuous relaxation training of discrete latent variable image models." Beysian DeepLearning workshop, NIPS. Vol. 201. 2017.

Singh, Gautam, Fei Deng, and Sungjin Ahn. "Illiterate dall-e learns to compose." arXiv preprint arXiv:2110.11405 (2021).

Williams, Will, Sam Ringer, Tom Ash, David MacLeod, Jamie Dougherty, and John Hughes. "Hierarchical quantized autoencoders." Advances in Neural Information Processing Systems 33 (2020): 4524-4535.

Alemi, Alexander, Ben Poole, Ian Fischer, Joshua Dillon, Rif A. Saurous, and Kevin Murphy. "Fixing a broken ELBO." In International conference on machine learning, pp. 159-168. PMLR, 2018.

---

> ### Author Response · Authors · 2023-05-23
>
> Thank you for your valuable comments.
>
> > (W1) The current work is mostly about an extended architecture of previous work (SQ-VAE), and the hierarchical architecture itself is directly adapted from literature (ResNet-VAE, VQ-VAE2), thus there is very little methodological contribution from a machine learning perspective.
>
> We first emphasize that SQ-VAE-2 and RSQ-VAE are two instantiations of the proposed general framework HQ-VAE, and HQ-VAE is not restricted to these specific structures. Our methodological contributions are as follows.
>
> (1)	We proposed a general formulation that can deal with the independently proposed VQ-VAE extensions, i.e., VQ-VAE-2 and RQ-VAE, in a unified way.
>
> (2)	We instantiated the variants of VQ-VAE models into the *top-down* layers, which allows flexible and easy design of the hierarchical structure.
>
> (3)	We provided a recipe for modeling hierarchical discrete representation from the construction of hierarchical latent structures to the derivation of objective functions to learn the model, with the demonstration of the hybrid model (see Appendix C.3).
>
> We believe that the unification and generalization of the independently proposed models are not trivial at all and there is no paper discussing the methodological connection between VQ-VAE-2 and RQ-VAE to our best knowledge. We achieved this in a theoretically solid manner. It is not straightforward to establish variational Bayes framework for such hierarchical VQ-VAE models. Furthermore, we instantiated the variants of VQ-VAE into the *top-down* layers, which should make the construction of the hierarchical structure easier and more flexible.
>
> We will clarify the methodological contributions and their benefits more in the manuscript.
>
>
> ---
>
> > (W2) The proposed advantages of HQ-VAE are mostly a rehash of those of SQ-VAE: ("(1) greatly reduces the number of hyperparameters to be tuned (only the one from the Gumbel-softmax trick), and (2) enhances codebook usage without any heuristics thanks to the self-annealing effect"). Given that the present work builds on SQ-VAE, it is insufficient to just demonstrate the benefit of SQ-VAE (and compare with the deterministically quantized VQ-VAE and variants), but it should rather focus on what advantages a hierarchical extension brings in addition to SQ-VAE.
>
> We note that our aim is not to improve SQ-VAE by incorporating hierarchical structure into the discrete latent or to discuss which type of latent structure leads to the best performance. Instead, our motivation is to provide general methodology to avoid the collapse issues and improve the reconstruction performance of the advanced variants of VQ-VAE model. We achieved this by unifying and formulating them in the variational Bayes framework.
>
> Which model between VQ-VAE/SQ-VAE, RQ-VAE/RSQ-VAE, and VQ-VAE-2/SQ-VAE-2 is an appropriate choice depends highly on applications and use-cases. Hierarchical discrete representation has been shown to have benefits against VQ-VAE if the model could be successfully trained. VQ-VAE-2 learns multi-resolution discrete representation, which enables to model global and local features with top and bottom layers, respectively (Razavi+2019). Such progressive reconstruction property is beneficial for generation tasks because one can divide the burden of the prior training into multiple layers (Razavi+2019, Dhariwal+2020). In contrast, RQ-VAE allows us to accommodate different compression rates only by changing the number of layers used during inference (Zeghidour+2021). This property is suitable for the application to neural codec (please refer to our reply posted in the thread “To all reviewers” for more detailed discussion about the application). SQ-VAE-2 and RSQ-VAE respectively inherit the properties of VQ-VAE-2 and RQ-VAE because they share the same latent structure. We will add the detailed explanation of the advantages of the hierarchical structures to (new) Section 6.2.
>
> Our proposal has a potential to improve many existing methods based the variants of VQ-VAE only by following the proposed modification of the training schemes. Furthermore, thanks to the generality of the formulation, advanced VQ-VAE models that will be proposed in the future may also benefit from our framework.
>
> [Razavi+2019] “Generating Diverse High-Fidelity Images with VQ-VAE-2”
>
> [Dhariwal+2020] “Jukebox: A Generative Model for Music”
>
> [Zeghidour+2021] “SoundStream: An end-to-end neural audio codec”

---

> > ### Author Response · Authors · 2023-05-23
> >
> > > (W3) If the focus was rather on the benefit of the probabilistic quantization aspect of HQ-VAE, then there is insufficient justification given there are existing probabilistic VQ-VAE formulations which are closely related to (or even subsume) HQ-VAE/SQ-VAE, such as Sonderby 2017 and discrete VAE (dVAE) (Singh 2022). A more detailed discussion and comparison would be needed.
> >
> > Thank you for the suggestion. We believe a comparison of SQ-VAE with the existing probabilistic discrete latent models especially from the perspective of hierarchization provides an insight into the benefits of our proposal.
> >
> > In the formulation of both of [Sønderby+2017] and [Ramesh+2020, Singh+2022] (dVAE), the categorical posterior distribution in the index space is directly modeled by the output of encoder. These models can mimic the stochastic quantization procedure and be trained by ELBO-based objective functions. However, such index-domain modeling cannot incorporate the codebook geometry explicitly into the posterior distribution. In particular, this approach cannot be extended to residual quantization scheme whereas VQ/SQ-based approach can assign code vectors that resemble the quantization residuals in a Euclidean sense. To summarize, VQ/SQ-based modeling allows such vector operations for the posterior modeling. On another note, in VQ/SQ-VAE, we can evaluate the reconstruction errors coming from the discretization with the quantization errors (please refer to [Dhariwal+2020]). We will add the above discussion to Section 2.2.
> >
> > [Dhariwal+2020] “Jukebox: A Generative Model for Music”
> >
> >
> > ---
> >
> > > (W4) The methodology for comparing the models seems problematic. The comparison based on reconstruction error / latent perplexity can be misleading and inclusive; it's more principled to compare rate-distortion curves like in Williams 2020. Having lower reconstruction error AND higher latent perplexity does not necessarily mean a better model; a VAE can in theory be tuned to operate anywhere on the R-D curve such that it operates at lower distortion and higher rate (Alemi 2018). Thus the conclusion that the HQ-VAE attains lower distortion / higher latent perplexity does not necessarily mean it's "better" than its deterministically quantized counterpart.
> >
> > Figures 2 (a-b) and 3 (a-b) correspond to such R-D curves.
> >
> > First, we would like to clarify the relationship of R-D curves between in [Alemi+2018] and [Williams+2020]. The application of this definition of the rate in [Alemi+2018] to discrete latent case leads to that the rate depends only on the number of bits (total capacity) in the latent. In [Alemi+2018], the rate is defined as an averaged KL divergence between posterior and prior distributions, which is an upper bound of mutual information of data samples and latent variables given by the encoder. This definition is applicable to the discrete latent cases as well. Note that, in both VQ-VAE and SQ-VAE cases, the prior is set to be a uniform categorical distribution and the posterior is always delta function during inference. In these cases, the rate becomes the logarithm of the total capacity in the latent, i.e., $d_z\log_2K$. That is why the rate in Figure 2 of [Williams+2020] is defined as the total capacity. Because x-axes of our plots (Figures 2 (a-b) and 3 (a-b)) indicate codebook size $K$ or the number of layers $d_l$ (determining $d_z$), they are based on such R-D curves.
> >
> > We agree that higher perplexity, which ranges from 1 to the number of codes, does not necessarily mean better performance. We would like to clarify the motivation why we introduced the metric. We used the metric in Table 1 and Figure 3 (c). Perplexity measures how the given codebook elements are evenly used, i.e., how the aggregated posterior is close to a uniform distribution in KL divergence. Since extreme small perplexity value means that the usage of codebook is extremely biased, we can use the metric to check if the issue of layer collapse occurred. According to Table 1, VQ-VAE-2 struggles with using the top layers efficiently (i.e., suffers from layer collapse).

---

> > > ### Author Response · Authors · 2023-05-26
> > > **Manuscript revision**
> > >
> > > We thank you again for your constructive feedback. We have revised the manuscript accordingly. We highlighted the revised parts in red.
> > >
> > > > (RC1) The paper needs to be more clear about its scope of contribution, given it's a straightforward extension of SQ-VAE and does not make significant methodological contribution. The claim to novelty -- "we propose a novel framework to stochastically learn hierarchical discrete representation on the basis of the variational Bayes framework" -- is thus problematic, as everything except the "hierarchical" part is existing work (SQ-VAE), and even the "hierarchical" part is fairly standard from literature.
> > >
> > > We have added modifications to Section 1 and the abstract. Please also refer to our comments on Weaknesses 1 and 2.
> > >
> > > ---
> > >
> > > > (RC2) Additionally, a more careful comparison of SQ-VAE with the existing probabilistic formulations of Sønderby 2017 and Singh 2022 would be insightful and add to the contributions, given the original SQ-VAE paper did not address these two overlapping prior works.
> > >
> > > We have reflected the suggestion on Section 2.2 by creating a new paragraph. Please also refer to our comments on Weakness 3.
> > >
> > > ---
> > >
> > > > (R3) The empirical comparison should also be based on R-D curves (training models with different weightings between rate and distortion losses) to be correct.
> > >
> > > We have emphasized that our plots are based on R-D curves in Section 5. Please also refer to our comments on Weakness 4.

---

### Review · Reviewer_hz6B · 2023-05-15

**Summary Of Contributions:**

The paper propose a hierarchical VAE with discrete latent variables. It adopts similar design of top-down and bottom-up paths of hierarchical VAEs with continuous latent variables as in previous work. The objective is ELBO, where the variational part comes from the stochastic quantization/de-quantization process. The paper compares HQ-VAE with previous discrete VAEs, and show stronger performance in reconstruction loss and perceptual quality of reconstructed image and higher perplexity for better use the codebook.

**Audience:**

Yes

**Claims And Evidence:**

Yes

**Requested Changes:**

As discussed in the weakness section, more in-depth study on the application is needed. I do not quite see the particular value of having a discrete VAE that does better in reconstruction. If the claim is that the latent space is more suitable for training a generative model, then an comprehensive quantitative comparison is needed, against generative model trained on VQ-VAE2 and single layer VQGAN. The latter is particularly interesting, as models relied on single layer VQ-GAN such as latent diffusion works particularly well. I am suspicious that hierarchical latent structure, while improving reconstruction,  makes training generative models harder.

**Strengths And Weaknesses:**

Strengths:

The paper is excellently written, and it presents a clear explanation of the model design using detailed formulas and figures. The figures for explaining the model structure is informative. Moreover, it offers a comprehensive overview of the existing literature on discrete VAEs and conducts a thorough comparison with other models. The proposed design is novel. As hierarchical VAEs with continuous latent variables have already demonstrated significant success, the development of a discrete counterpart trained with ELBO is imperative. Finally, the experimental results showcase some improvements over prior models, further validating the efficacy of the proposed approach.

Weakness:

The application of this model seems to be limited to reconstruction. Although it shows better reconstruction compared to baselines on a variety of dataset, reconstruction itself is not a very meaningful application. The application is particularly important for hierarchical VAE because the latent space itself is of large size so we can't really claim it does well in data compression, which is one of the important application of auto-encoder. The authors needs to justify what is the real benefit of designing a hierarchical auto-encoder that can reconstruct data well. In the appendix, the author show that it can be used to train generative model on latent variables. However, only some qualitative results are shown, without comparing with the same model trained on the latent space of other VAEs.

---

> ### Author Response · Authors · 2023-05-23
>
> Thank you for your valuable comments.
>
> > (W1) The application of this model seems to be limited to reconstruction. Although it shows better reconstruction compared to baselines on a variety of dataset, reconstruction itself is not a very meaningful application. … The authors needs to justify what is the real benefit of designing a hierarchical auto-encoder that can reconstruct data well.
> > I do not quite see the particular value of having a discrete VAE that does better in reconstruction.
>
> **Benefits of hierarchical structure**
>
> Regarding how the hierarchical structures of RSQ-VAE and SQ-VAE-2 are useful for practical applications, please refer to the thread “To all reviewers”. Generally, hierarchical representation learning has been shown to be able to capture discrete latent features in progressive ways, which is beneficial not only for codec (Zeghidour+2021, Défossez+2022) but also for generation tasks (Razavi+2019, Shu+2022) and downstream tasks (Castellon+2021). We empirically confirmed that our hierarchical models have this property as shown in Figures 4 (a) and (b).
>
> [Razavi+2019] “Generating Diverse High-Fidelity Images with VQ-VAE-2”
>
> [Zeghidour+2021] “SoundStream: An end-to-end neural audio codec”
>
> [Castellon+2021] “Codified audio language modeling learns useful representations for music information retrieval”
>
> [Défossez+2022] “High fidelity neural audio compression”
>
> [Shu+2022] “Bit prioritization in variational autoencoders via progressive coding”
>
> **Benefits of obtaining hierarchical model that can reconstruct data well**
>
> VAEs including the VQ-VAE family have a trade-off between compression (rate) and reconstruction accuracy (distortion) (Alemi+2018, Wiliams+2020). [Alemi+2018] stated that a proper way to assess the performance of VAE-based representation learning is to measure the mutual information between data and latent spaces, which can be expressed by using the rate—distortion (RD) curve from information theory. We compared our HQ-VAEs with the baselines based on such curves (e.g., Figures 2 and 3). Furthermore, we introduced multiple metrics for reconstruction accuracy including both Euclidean and perceptual ones, and confirmed that HQ-VAEs obtain better RD curves than the baselines in all the metrics. The results suggest us two benefits of HQ-VAE as follows.
>
> (B1) HQ-VAEs outperform the baselines in terms of reconstruction accuracy with the same latent capacities.
>
> (B2) HQ-VAEs achieve the comparable reconstruction performance as the baselines with the higher compression rate, i.e., the smaller latent capacity (e.g., smaller codebook size $K$, fewer layers $L$).
>
> As for generation tasks, it is generally easier for prior models to learn the distribution in the discrete latent space with the lower latent capacity (Rombach+2022). In contrast, lower latent capacity usually leads to worse reconstruction performance due to the RD trade-off. The lower reconstruction performance deteriorates generation performance because the reconstruction performance caps the quality of generated samples. Thus, metrics for reconstruction accuracy provide an estimate on the achievable generation performance (Esser+2021). Accordingly, the second benefit (B2) is crucial even for the application to generation tasks.
>
> We will explain the above in the revised manuscript to clarify the benefit of obtaining hierarchical models that can reconstruct data well.
>
> [Alemi+2018] “Fixing a broken ELBO”
>
> [Williams+2020] “Hierarchical quantized autoencoders”
>
> [Esser+2021] “Taming transformers for high-resolution image synthesis”
>
> [Rombach+2022] “High-Resolution Image Synthesis with Latent Diffusion Models”
>
>
> ---
>
> > (W2) The application is particularly important for hierarchical VAE because the latent space itself is of large size so we can't really claim it does well in data compression, which is one of the important application of auto-encoder.
>
> We agree that discussing potential applications is important for future studies. Then, we summarized the possible application of HQ-VAE while considering the benefits of hierarchical latent structure. Please refer to our reply posted in the thread “To all reviewers”.
>
> In addition, we would like to note that hierarchical VAE does not necessarily have a large latent capacity. One can flexibly set the latent capacity of VQ-VAE (e.g., by changing the codebook size $K$ and resolution of each discrete tensor), regardless of whether the model is hierarchical or not. In other words, given a target compression rate, one can design both SQ-VAE-2 and RSQ-VAE with the target rate. Furthermore, RSQ-VAE enables us to change the compression rate in the inference phase even after the model is trained as mentioned in the thread “To all reviewers”.

---

> > ### Author Response · Authors · 2023-05-23
> >
> > > (W3) … models relied on single layer VQ-GAN such as latent diffusion works particularly well. I am suspicious that hierarchical latent structure, while improving reconstruction, makes training generative models harder.
> > > In the appendix, the author show that it can be used to train generative model on latent variables. However, only some qualitative results are shown, without comparing with the same model trained on the latent space of other VAEs
> >
> > [Razavi+2019, Lee+2022a] suggested that incorporating hierarchical structure into the discrete latent makes the prior training easier. It is mainly because one can reduce the length of sequence at the top level latent layer thanks to the hierarchical structure. Regarding the priors for from the second to the bottom layers, one can make the use of the strong correlation between the layers and the previous layers. For example, thanks to such hierarchical structure, RQ-VAE outperforms VQ-GAN in terms of generation performance even with the smaller architecture size of prior models (refer to Tables 2 of [Lee+2022a]) despite that the reconstruction performance of RQ-VAE is slightly worse than that of VQ-GAN (refer to Table 4 of [Lee+2022a]). The experimental results support that the hierarchical structure is beneficial even to generation tasks. As a side note, the application of diffusion model to prior modeling for the hierarchical models is also an interesting future direction. However, the selection of prior models is orthogonal to our contributions.
> >
> > In response to the comment on the experiments in the appendix, we conducted additional experiments for the generation task on FFHQ dataset. First, we calculated FID score of our generative model presented in Appendix D to numerically evaluate the generation performance. The obtained FID score is 8.46, which is the state-of-the-art score among VAE- and VQ-VAE-based models including very deep VAE (Child2021), VQ-GAN (Esser+2021) and RQ-VAE (Lee+2022a). Next, for a fair comparison with RQ-VAE model (Lee+2022a), we trained RQ-Transformer (Lee+2022a) on the latent space of our RSQ-VAE model. The FID score of our RSQ-VAE+RQ-Transformer is 9.74, which is still a better score than that of RQ-VAE+RQ-Transformer. The difference between our RSQ-VAE+RQ-Transformer and RQ-VAE+RQ-Transformer is only in the training schemes of the VAE models. Still, RSQ-VAE leads to better generation performance than RQ-VAE even without adversarial training. We should also note that all of VQ-GAN, RQ-VAE, and RSQ-VAE in this experiment are implemented with almost the same encoder—decoder architecture and have the same latent capacity (2816 bits). We will add the numerical results to the current manuscript to Appendix D.
> >
> > |  | FID |
> > | ---- | ---- |
> > | Very Deep VAE (Child2021) | 28.5 |
> > | VQ-GAN + Transformer (Esser+2021) | 11.4 |
> > | RQ-VAE + RQ-Transformer (Lee+2022a, SOTA) | 10.38 |
> > | **RSQ-VAE** + RQ-Transformer | **9.74** |
> > | **RSQ-VAE** + contextual RQ-Transformer | **8.46** |
> >
> > [Razavi+2019] “Generating Diverse High-Fidelity Images with VQ-VAE-2”
> >
> > [Esser+2021] “Taming transformers for high-resolution image synthesis”
> >
> > [Child2021] “Very deep VAEs generalize autoregressive models and can outperform them on images”
> >
> > [Lee+2022a] “Autoregressive image generation using residual quantization”

---

> > > ### Author Response · Authors · 2023-05-26
> > > **Manuscript revision**
> > >
> > > We thank you again for your constructive feedback. We have revised the manuscript accordingly. We highlighted the revised parts in red.
> > >
> > > - We have reflected our replies to (W1) and (W2) on Section 6.2.
> > >
> > > - We have added the numerical results reported in our comment on (W3) to Appendix D.

---

### Author Response · Authors · 2023-05-23
**To all reviewers**

We sincerely appreciate your valuable and constructive feedback.

We summarized our takeaway messages of this work below while conducting an additional experiment.

**Empirical comparisons of SQ-VAE-2 and RSQ-VAE**

We conducted an additional experiment to compare reconstruction performance of SQ-VAE-2 and RSQ-VAE. We compared them in three different compression rates by changing the number of code vectors. Interestingly, SQ-VAE-2 achieves better reconstruction performance in the case of the lower compression rate, whereas RSQ-VAE reconstructs the original images better than SQ-VAE-2 in the case of higher compression rate. We will add the experimental result to Section 5.3.

**Takeaway messages**

First, many applications with VQ-VAE-2 and RQ-VAE have been proposed in various domains so far. In these applications, VQ-VAE-2 and RQ-VAE can be basically replaced with SQ-VAE-2 and RSQ-VAE for improvements in terms of efficient codebook usage and reconstruction accuracy. Our approach eliminates the need for the repetition of tuning many hyper-parameters and the introduction of ad-hoc techniques.

In common with SQ-VAE-2 and RSQ-VAE, both are applicable to generative modeling with the additional training of prior models as in a bunch of previous work. Recently, RQ-VAE is more often used for the generation tasks than VQ-VAE-2 due to the severe instability issue of VQ-VAE-2, i.e., layer collapse (Dhariwal+2020). However, we believe that the proposal of SQ-VAE-2 has a potential to advocate the use of such hierarchical model for generation tasks since it mitigates the issue greatly. In contrast, SQ-VAE-2 and RSQ-VAE have their own unique advantages as below.

SQ-VAE-2 tested in the experiment was shown to be able to learn multi-resolution discrete representation thanks to the design of *bottom-up* path with the pooling operators (see Figure 4(a)). The results implied that further semantic disentanglement of the discrete representation might be possible by adopting specific inductive architectural components in the *bottom-up* path, which is interesting future direction. As a second note, in the case of lower compression rates, SQ-VAE-2 outperforms RSQ-VAE in terms of reconstruction performance according to the additional experiment. Hence, SQ-VAE-2 might be appropriate for high-fidelity generation tasks when the prior model could be larger as in [Razavi+2019].

One of the strengths of RSQ-VAE (and RQ-VAE) is that one can easily accommodate different compression rates only by changing the number of layers during the inference i.e., without changing or retraining the model (Zeghidour+2021). Furthermore, in the case of higher compression rates, RSQ-VAE outperforms SQ-VAE-2 in reconstruction performance. The properties are suitable for the application of neural codec as in [Zeghidour+2021, Défossez+2022].

Furthermore, we also formulate the hybrid model to demonstrate that one can flexibly combine the different types of *top-down* layer. It implies the potential of using the different types of the layers to disentangle the information, which is also interesting future direction.

[Razavi+2019] “Generating diverse high-fidelity images with VQ-VAE-2”

[Dhariwal+2020] “Jukebox: A generative model for music”

[Zeghidour+2021] “SoundStream: An end-to-end neural audio codec”

[Défossez+2022] “High Fidelity Neural Audio Compression”

---

### Decision · Action_Editors · 2023-06-25

**Recommendation:** Reject

**Comment:**

The reviewers praised the direction the paper is taking in the context of deep generative models. However, they also raised several concerns before and after the discussion phase.

First, there is no clear-cut evaluation of HQ-VAEs as generative models for sampling and for representation learning. While the paper claims are all about reconstruction errors, the lack of extensive experiments for these other two perspectives leaves some big open questions about the necessity of the framework. The variants of the RD-curves provided do not help understand where these models stand in these scenarios. The authors provided some preliminary qualitative experiments in the appendix and added more experiments on FFHQ for RSQ-VAE in the rebuttal. I agree with the reviewers that authors should perform a complete analysis on the generative and representation learning aspects, including all architectures and datasets. This is currently the major missing piece towards acceptance.

Second, while extensive (at least from the point of view of reconstruction), the experiments do not provide error-bars nor statistical tests, and therefore it is less clear whether the reported gains are substantial. In the added experiments on the MUSHRA listening test authors are showing distributions over the obtained scores. The authors promised to add the error bars to Figs 2-3, which are very hard to see.  It is not clear how were they computed and on how many trials.

Third, the comparison and discussion w.r.t. previous architectures and AE design such as SQ-VAE and dVAE is preliminary.
One reviewer claims that there is not enough novelity in extending the stochasticity to hierarchical AEs. I disagree, as this is not one aspect papers should be evaluated at TMLR. However, I recognize that the paragraph added by the authors in Sec 2.2 can be more precise and further expanded to discuss the similarities between SQ-VAEs and HQ-VAEs via residual-enanched AEs as the reviewer points out. This will greatly benefit readers properly understanding and navigating the hierarchical AE landscape.

Additionally, one reviewer questions the deeper motivation of using discrete representations in the first place. I disagree in that discrete codes have a clear advantage in terms of compression and interpretability, and while it would be interested to include also only-continuous latent variable models, this would go outside the scope of the paper.

The authors engaged in the discussion but only partially addressed the above concerns. Two reviewers voted for rejection. I agree that the manuscript in its current status (despite having improved a lot through the discussion) is not ready for publication. The current status would be that of a major revision, which is unfortunately not possible on TMLR. Authors are encouraged to address the above points, especially completing the experiments on generative modeling and representation learning, and resubmit.

**Audience:**

The topic is definitely of interest for the ML and deep generative modeling audience of TMLR.

**Claims And Evidence:**

The paper deals with learning hierarchical stochastic autoencoders (AEs) with discrete latent representations via variational Bayes. Specifically, the authors propose a general framework that extends several previous architectures such as VQ-VAE2 (which is a deterministic AE) and RQ-VAE into a unified representation that is amenable to be trained by maximising an ELBO.  This is in line with other recently proposed stochastically-quantized AEs SQ-VAE such as Takida et al. 2022.

In their model, named HQ-VAE, they use bidrectional inference and empirically showcase that this allows to lower the reconstruction error of the proposed hierarchical models, which surpass previous architectures both in terms of Euclidean and perceptual distance. At the same time, it also improves w.r.t. the entropy of the learned discrete codebook. However, HQ-VAEs are not evaluated for other tasks that one latent-variable generative model is expected to support: sample generation and representation learning.

**Resubmission Of Major Revision:**

The authors may consider submitting a major revision at a later time.